# WHEN SHOULD WE PREFER DECISION TRANSFORMERS FOR OFFLINE REINFORCEMENT LEARNING?

**Prajjwal Bhargava**[1], **Rohan Chitnis**[1], **Alborz Geramifard**[1], **Shagun Sodhani**[1], **Amy Zhang**[1,2]

[1]Meta AI    [2] University of Texas, Austin

{prajj, ronuchit, alborzg, sodhani, amyzhang}@meta.com

## ABSTRACT

Offline reinforcement learning (RL) allows agents to learn effective, return-maximizing policies from a static dataset. Three popular algorithms for offline RL are Conservative Q-Learning (CQL), Behavior Cloning (BC), and Decision Transformer (DT), from the class of Q-Learning, Imitation Learning, and Sequence Modeling respectively. A key open question is: which algorithm is preferred under what conditions? We study this question empirically by exploring the performance of these algorithms across the commonly used D4RL and ROBOMIMIC benchmarks. We design targeted experiments to understand their behavior concerning data suboptimality, task complexity, and stochasticity. Our key findings are: (1) DT requires more data than CQL to learn competitive policies but is more robust; (2) DT is a substantially better choice than both CQL and BC in sparse-reward and low-quality data settings; (3) DT and BC are preferable as task horizon increases, or when data is obtained from human demonstrators; and (4) CQL excels in situations characterized by the combination of high stochasticity and low data quality. We also investigate architectural choices and scaling trends for DT on ATARI and D4RL and make design/scaling recommendations. We find that scaling the amount of data for DT by 5x gives a 2.5x average score improvement on ATARI.

## 1 INTRODUCTION

Offline reinforcement learning (RL) (Levine et al., 2020; Lange et al., 2012; Ernst et al., 2005) aims to leverage existing datasets of agent behavior in an environment to produce effective policies. A key open question is: *which learning method is preferred for offline RL?* In this paper, we empirically investigate this question. Among many offline RL algorithms, we focus on three that have been studied extensively and are thus relatively easy to interpret and compare: Conservative Q-Learning (CQL) (Kumar et al., 2020), Behavior Cloning (BC) (Bain and Sammut, 1995), and Decision Transformer (DT) (Chen et al., 2021). CQL, from the class of Q-Learning (Sutton and Barto, 2018), uses temporal difference (TD) updates to learn a value function through bootstrapping. In theory, it can learn effective policies even from highly suboptimal trajectories in stochastic environments, but in practice, it suffers from instability and sensitivity to hyperparameters in the offline setting (Brandfonbrener et al., 2022). BC, from the family of Imitation Learning (Hussein et al., 2017), mimics the behavior policy of the data; however, it relies on the data being high-quality. DT, from the class of Sequence Modeling, is a recently popularized paradigm that aims to transfer the success of Transformers (Vaswani et al., 2017) into offline RL (Chen et al., 2021; Janner et al., 2021), yet they have shown to struggle with stochastic dynamics (Brandfonbrener et al., 2022).

We design targeted experiments to understand how these three algorithms perform as we vary properties of the data, task, and environment. Our experiments are conducted across the commonly used D4RL, ROBOMIMIC, and ATARI benchmarks. Table 1 shows a high-level summary of our key findings. In Section 4.1, we begin by establishing baseline results in our benchmark tasks for CQL, BC, and DT, for both dense-reward and sparse-reward settings. Then, we perform experiments to answer several key questions, which form the core contributions of this paper:

- (Sections 4.2, 4.3, 4.4) How are agents affected by the presence of **suboptimal data**? As the notion of suboptimality can take on many meanings in offline RL, we consider three definitions:

| Algorithm | Property | | | | | | | |
|-----------|--------|--------|---------|--------|-------|---------|-------|--------------|
| | SPARSE | BEST X% | WORST X% | LENGTH | NOISE | HORIZON | SPACE | STOCHASTICITY |
| DT | ✓ | × | ✓ | × | ✓ | ✓ | × | × |
| CQL | × | ✓ | ✓ | × | ✓ | × | × | ✓ |
| BC | × | × | × | ✓ | × | ✓ | × | × |

Table 1: This paper compares the robustness of three algorithms for offline RL to various properties of the data, task, and environment: sparse rewards (SPARSE), training on the best X% of data (BEST X%), training on the worst X% of data (WORST X%), trajectory lengths in the dataset (LENGTH), presence of highly suboptimal actions in the dataset (NOISE), long task horizon (HORIZON), large task state space (SPACE) and Stochasticity (STOCHASTICITY). A check mark indicates relative robustness, while a cross indicates notable deterioration in performance. We see that each algorithm is preferable in different situations.

- (Section 4.2) Our first setting involves **varying the amount of data the agents are trained on**. More specifically, we sort trajectories in the dataset by their returns and expose the agents to either the best X% or the worst X% of the data for varying values of X. This enables us to study sample efficiency when learning from high-quality and low-quality data.
- (Section 4.3) In our second experiment, we study the impact of suboptimality arising from **increased trajectory lengths** in the dataset. In longer trajectories, rewarding states are typically further away from early states, which can impact training dynamics.
- (Section 4.4) Finally, in our third experiment, we examine the impact of **adding noise to data**, in the form of random actions. This setting can be seen as simulating a common practical situation where the offline dataset was accompanied by a lot of exploration.

- (Section 4.5) How do agents perform when the **task complexity** is increased? To understand this, we study the impact of both state space dimensionality and task horizon on agent performance.
- (Section 4.6) How do agents perform in **stochastic environments**? To investigate this, we evaluate the performance of agents as we vary the degree of stochasticity and data quality.
- (Sections 4.7) How can one **effectively use DT in practice**? Based on the overall strength of DT in our findings, we provide guidance on architecture (Appendix A) and hyperparameters for DT and perform a detailed analysis of scaling the model size and amount of data in ATARI.

Our key findings are: (1) DT is more robust than CQL but also requires more data; (2) DT is the best choice in sparse-reward and low-quality data settings; (3) DT and BC are preferable as task horizon increases, or when data is obtained from suboptimal human demonstrators; (4) CQL excels in situations characterized by the combination of high stochasticity and low data quality; (5) larger DT models require less training, and scaling the amount of data improves scores on ATARI.

Our work aligns with a recent research trend that examines the trade-offs among various algorithms for offline RL. Brandfonbrener et al. (2022) outlined theoretical conditions, such as nearly deterministic dynamics and prior knowledge of the conditioning function, under which Sequence Modeling (called "RCSL" in their work) is preferable. Our paper extends their research by asking carefully designed questions targeted to provide novel empirical insights. Kumar et al. (2023) investigated when Q-Learning might be favored over Imitation Learning. Our research expands on this inquiry by incorporating the recently popularized DT as part of the Sequence Modeling paradigm, thereby providing insight into the training dynamics and learned policies of each algorithm.

We hope this paper helps researchers determine which offline RL algorithm to use in their application. Throughout the paper, we provide practical guidance on which algorithm is preferred, given characteristics of the application. Code and data: https://github.com/facebookresearch/rl_paradigm.

## 2 RELATED WORK

Our work addresses the question of which learning algorithm is most suitable for offline RL (Levine et al., 2020; Fu et al., 2020; Prudencio et al., 2023) by examining three prominent algorithms from the following paradigms in the field: Q-Learning, Imitation Learning, and Sequence Modeling (Brandfonbrener et al., 2022). We chose CQL and DT as representative algorithms for Q-Learning and Sequence Modeling, respectively, based on their popularity in the literature and their strong performance across standard benchmarks (Kumar et al., 2020; 2023; Chen et al., 2021; Lee et al., 2022; Kumar et al.,

2023). Other options are possible as well such as BCQ (Fujimoto et al., 2019), BEAR (Kumar et al., 2019a), and IQL (Kostrikov et al., 2021). The Trajectory Transformer is another algorithm within the Sequence Modeling paradigm (Janner et al., 2021) (model-based). Similar to Sequence Modeling, other studies have explored approaches that aim to learn policies conditioned on state and reward (or return) to predict actions (Schmidhuber, 2019; Srivastava et al., 2019; Brandfonbrener et al., 2022; Kumar et al., 2019b; Emmons et al., 2022), in both online and offline settings. Finally, despite its simplicity, Behavior Cloning (BC) remains a widely used baseline among Imitation Learning algorithms (Kumar et al., 2023; Chen et al., 2021; Ho and Ermon, 2016; Fujimoto and Gu, 2021a), and thus was chosen as the representative algorithm from the Imitation Learning paradigm. Alternative options, such as TD3-BC (Fujimoto and Gu, 2021b), could also be considered.

While our research focuses on these three paradigms, it is also worth mentioning model-based RL approaches, which have begun to increase in popularity recently. These approaches have delivered promising results in various settings (Janner et al., 2022; Yu et al., 2020a; Kidambi et al., 2020a; Argenson and Dulac-Arnold, 2021), but we do not examine them in our work, instead choosing to focusing on the most prominent paradigms in offline RL (Tarasov et al., 2022).

In light of the recent interest in scaling foundational models (Hoffmann et al., 2022), both Kumar et al. (2023) and Lee et al. (2022) have demonstrated that DT scales with parameter size. Moreover, Kumar et al. (2023) indicate that CQL performs better on suboptimal dense data in the Atari domain. Our findings concur with these studies but offer a more comprehensive perspective, as we also explore sample efficiency, as well as the scaling of parameters and data together.

## 3 PRELIMINARIES

Here, we discuss brief background (see Appendix C for more details) and our experimental setup.

### 3.1 BACKGROUND

In reinforcement learning (RL), an agent interacts with a Markov decision process (MDP) (Puterman, 1990), taking actions that provide a reward and transition the state according to an unknown dynamics model. The agent's objective is to learn a policy that maximizes its return, the sum of expected rewards. In offline RL (Levine et al., 2020), agents cannot interact with the MDP, but instead learn from a fixed dataset of transitions $\mathcal{D} = \{(s, a, r, s')\}$, generated from an unknown behavior policy.

Q-Learning (Sutton and Barto, 2018) uses temporal difference (TD) updates to estimate the value of actions via bootstrapping. We focus on Conservative Q-Learning (CQL) (Kumar et al., 2020), which addresses overestimation by constraining Q-values so that they lower-bound the true value function. Behavior Cloning (BC) (Bain and Sammut, 1995) is a simple algorithm that mimics the behavior policy via supervised learning on $\mathcal{D}$. Sequence Modeling is a recently popularized class of offline RL algorithms that trains autoregressive models to map the trajectory history to the next action. We focus on the Decision Transformer (DT) (Chen et al., 2021), in which the learned policy produces action distributions conditioned on trajectory history and desired returns-to-go, $\hat{R}_{t'} = \sum_{t=t'}^{H} r_t$. DT is trained using supervised learning on the actions. Conditioning on returns-to-go enables DT to learn from suboptimal data and produce a wide range of behaviors during inference.

### 3.2 EXPERIMENTAL SETUP

**Data:** We consider tasks from two benchmarks: D4RL and ROBOMIMIC, chosen due to their popularity (Nie et al., 2022; Goo and Niekum, 2022). We also explore the HUMANOID GYM environment, which is not part of D4RL; in the process, we generate D4RL-style datasets for HUMANOID (details in Appendix D). We additionally conduct experiments in ATARI, which allows us to study the scaling properties of DT with image observations. All tasks are deterministic and fully observed (see Laidlaw et al. (2023) for a description of why deterministic MDPs are still challenging in RL) and have continuous state and action spaces, except for ATARI, which has discrete state and action spaces.

In **D4RL** (Fu et al., 2020), we focus on the HALFCHEETAH, HOPPER and WALKER tasks. For each task, three data splits are available: medium, medium replay, and medium-expert. These splits differ in size and quality. The medium split is obtained by early stopping the training of a SAC agent (Haarnoja et al., 2018) and collecting 1M samples from the partially trained behavior policy.

The medium replay split contains ∼100-200K samples, obtained by recording all interactions of the agent until it reaches a medium level of performance. The medium-expert split contains 2M samples obtained by concatenating expert demonstrations with medium data. In **ROBOMIMIC** (Mandlekar et al., 2021), we consider four tasks: Lift, Can, Square, and Transport. Each task requires a robot to manipulate objects into desired configurations; see Mandlekar et al. (2021) for details. For each task, three data splits are provided: proficient human (PH), multi-human (MH), and machine-generated (MG). PH data has 200 demonstrations collected by a single experienced teleoperator, while MH data has 300 demonstrations collected by teleoperators at varying proficiency levels. MG data has 300 trajectories, obtained by rolling out various checkpoints along the training of a SAC agent, and has a mixture of expert and suboptimal data. Appendix E gives details on ATARI (Agarwal et al., 2020).

**Evaluation Metrics:** On D4RL, we evaluate agents on normalized average returns, following Fu et al. (2020). On ROBOMIMIC, we measure success rates following Mandlekar et al. (2021). ATARI scores were normalized following Hafner et al. (2021). Scores are averaged over 100 evaluation episodes for D4RL and ATARI, and 50 for ROBOMIMIC. *All experiments report the mean score and standard deviation over five independent seeds of training and evaluation.*

**Details:** For DT, we used a context length of 20 for D4RL and 1 for ROBOMIMIC; see Section A for experiments and discussion on how context length impacts DT. All agents have fewer than 2.1M parameters (for instance, in D4RL, we have the following parameter counts: BC = 77.4k, CQL = 1.1M, DT = 730k). BC and CQL use MLP architectures. For more details, see Appendix H.

## 4 EXPERIMENTS

### 4.1 ESTABLISHING BASELINE RESULTS

We start by analyzing the baseline performance of the three agents on each dataset since we will make alterations to the data in subsequent experiments. We studied both the sparse-reward and the dense-reward regimes in D4RL and ROBOMIMIC. Because D4RL datasets have dense rewards only, we created the sparse-reward versions by exposing the sum of rewards in each trajectory only on the last timestep. For ROBOMIMIC experiments, we used the MG dataset (Section 3.2), which contains both sparse-reward and dense-reward splits. See Table 2 and Table 3 for results.

We noticed three key trends. (1) DT was generally better than or comparable to CQL in the dense-reward regime. For example, on ROBOMIMIC, DT outperformed CQL and BC by 154% and 51%, respectively. However, DT performed about 8% worse than CQL on D4RL. (2) DT was quite robust to reward sparsification, outperforming CQL and BC by 88% and 98% respectively on D4RL, and 194% and 62% respectively on ROBOMIMIC. These results are especially interesting because the dense and sparse settings have the *same states and actions* in the dataset; simply redistributing reward to the last step of each trajectory caused CQL's performance to reduce by half in the case of D4RL, while DT's performance remained unchanged. A likely reason is that sparser rewards mean CQL must propagate TD errors over more timesteps to learn effectively, while DT conditions on the returns-to-go at each state, and so is less affected by reward redistribution. (3) BC was never competitive with the best agent due to the suboptimality of the data collection policy.

We note that our methodology behind making D4RL tasks sparse is the same as (Chen et al., 2021) and might break Markovian dynamics. We provide an additional data point on the Maze2D environment, which provides sparse and dense rewards following Markovian dynamics, in Table 6 in Appendix G.

| Dataset | DT | | CQL | | BC |
|---|---|---|---|---|---|
| | Sparse | Dense | Sparse | Dense | |
| medium | **62.56 ± 1.16** | 63.66 ± 0.55 | 43.94 ± 4.7 | **67.11 ± 0.24** | 53.91 ± 5.93 |
| medium replay | **64.08 ± 1.25** | 65.22 ± 1.57 | 49.04 ± 13.79 | **78.41 ± 0.45** | 14.6 ± 8.32 |
| medium expert | **103.15 ± 0.77** | 103.64 ± 0.12 | 29.36 ± 5.14 | **105.39 ± 0.84** | 47.72 ± 5.5 |
| Average | **76.6 ± 1** | 77.51 ± 1.12 | 40.78 ± 7.88 | **83.64 ±0.51** | 38.74 ± 6.58 |

Table 2: Results on D4RL in both sparse and dense reward settings, averaged over the HALFCHEETAH, HOPPER and WALKER tasks. BC is reward-agnostic. DT performed best with sparse rewards, while CQL performed best with dense rewards. For full results, refer to Table 12 in Appendix G.

| Dataset | DT | | CQL | | BC |
|---|---|---|---|---|---|
| | Sparse | Dense | Sparse | Dense | |
| Lift | **93.2 ± 3.2** | **96 ± 1.2** | 60 ± 13.2 | 68.4 ± 6.2 | 59.2 ± 6.19 |
| Can | **83.2 ± 0** | **83.2 ± 1.6** | 0 ± 0 | 2 ± 1.2 | 55.2 ± 5.8 |
| Average | **88.2 ± 1.6** | **89.6 ± 1.4** | 30 ± 6.6 | 35.2 ± 3.7 | 57.2 ± 6 |

Table 3: Results on the MG dataset for two tasks (Lift and Can) in ROBOMIMIC. We see that DT performed best in both sparse and dense reward settings. Results on the Square and Transport task can be found in Table 13 and Figure 3 (Task Horizon) respectively.

> **Practical Takeaway:** Although CQL excels in certain dense-reward settings, its performance is subject to volatility in other environments. CQL is less preferable for use in sparse-reward settings. Meanwhile, DT is a competitive and risk-averse option that performs well in dense-reward settings and stands out as the top-performing agent in sparse-reward settings.

## 4.2 HOW DOES THE AMOUNT AND QUALITY OF DATA AFFECT EACH AGENT'S PERFORMANCE?

In this section, we aim to understand how agent performance varies given various amounts of high-return and low-return dense-reward data. To do this, we sorted the trajectories based on their returns and trained the agents on the "best" and "worst" X% of the data for various values of X. Analyzing each agent's performance on the best X% enables us to understand sample efficiency: how quickly do agents learn from high-return trajectories? Analyzing the performance on the worst X% enables us to understand how well the agents learn from low-quality data. See Figure 1 for results on D4RL.

The "-best" curves show that both CQL and DT improved as they observed more high-return data, but CQL was more sample-efficient, reaching its highest score at ~5% of the dataset, while DT required ~20%. We hypothesize that CQL is best in the low-data regime because in this case, the behavior policy is closer to the optimal one. However, as evidenced by the "CQL-best" line in Figure 1a going down between 20% and 80%, adding lower-return data could sometimes *harm* CQL, perhaps because (1) the difference between the optimal policy and the behavior policy grew larger and (2) TD updates had less opportunity to propagate values for high-return states. Meanwhile, DT was more stable and never worsened with more data, likely because conditioning on returns-to-go allowed it to differentiate trajectories of varying quality. BC's performance was best with a small amount of high-return data and then deteriorated greatly, which is expected because BC requires expert data.

The "-worst" curves show that DT learned from bad data 33% faster on average than CQL in medium replay (Figure 1a), but that they performed similarly in medium expert (Figure 1b). This is sensible because the low-return trajectories in medium replay are far worse than those in medium expert, and we have already seen that DT is more stable than CQL when the behavior policy is more suboptimal.

In Figure 10 in Appendix G.2, we show the same graphs as Figure 1 but for the *sparse-reward* D4RL dataset. This experiment revealed two new takeaways: (1) DT-best is far more sample-efficient and performant than CQL-best when rewards are sparse; (2) suboptimal data plays a more critical role for CQL in the sparse-reward setting than in the dense-reward setting.

> **Practical Takeaways:** 1) CQL is the most sample-efficient agent given a small amount of high-quality data; 2) While DT requires more data than CQL, it scales more robustly with additional suboptimal data due to DT's reliance on returns-to-go being more robust to variance in rewards; 3) DT can be slightly better than CQL with very low-quality data; 4) DT and CQL are preferable over BC, especially in the presence of suboptimal data.

## 4.3 HOW ARE AGENTS AFFECTED WHEN TRAJECTORY LENGTHS IN THE DATASET INCREASE?

In this section, we study how performance varies as a function of trajectory lengths in the dataset; this is an important question because, in practice, data suboptimality often manifests as longer demonstrations. To study this question, we turned to *human* demonstrations, which are qualitatively different from synthetic data (Orsini et al., 2021): human behavior is multi-modal and may be non-Markovian, so humans exhibit greater diversity in demonstration lengths when solving a task.

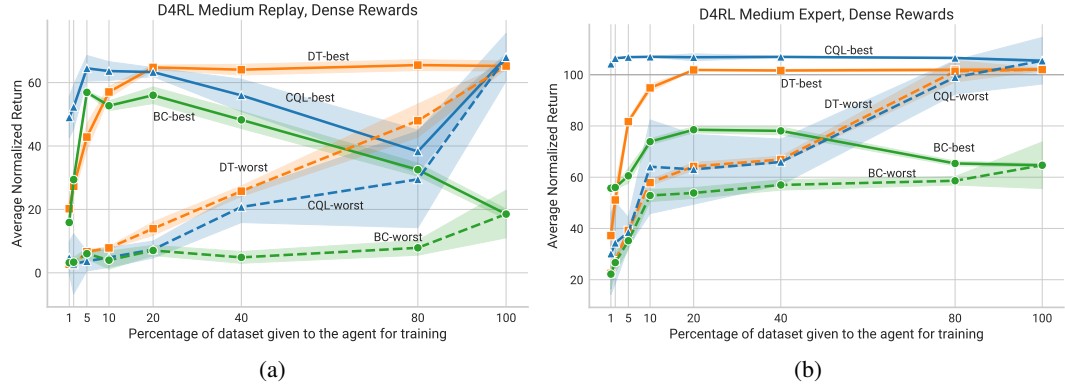

Figure 1: Normalized D4RL returns obtained by training DT, CQL, and BC on various amounts of highest-return ("best") or lowest-return ("worst") data. The left plot (a) is on medium replay data, while the right plot (b) is on medium expert data; both plots average over the HALFCHEETAH, HOPPER and WALKER tasks. Notice that the Y-axis limits are set lower on the left plot. We observe that CQL was the most sample-efficient agent when only a small amount of high-quality data was available, but it could degrade with lower-quality data. Meanwhile, the performance of DT never worsened with more data. Also, BC performed best with 20-40% of high-quality data, highlighting the importance of expert data for BC. For full results, refer to Figure 9 in Appendix G.

We used the ROBOMIMIC benchmark, which contains PH (proficient human) and MH (multi-human) sparse-reward datasets (Section 3.2). Because the rewards are fixed and given at the end of trajectories, the lengths of the trajectories are a proxy for optimality, as highlighted by Mandlekar et al. (2021). The MH datasets are further broken into "Better," "Okay," and "Worse" splits based on the proficiency of the demonstrator. We leveraged this to do more granular experimentation. See Table 4 for results.

| Dataset Type | Training Trajectory Length | DT | CQL | BC |
|---|---|---|---|---|
| PH | $105 \pm 13$ | $83.1 \pm 0.8$ | $45.6 \pm 5.0$ | $\mathbf{91.3 \pm 0.9}$ |
| MH-Better | $133 \pm 33$ | $53.5 \pm 0.6$ | $36.5 \pm 4.7$ | $\mathbf{80.2 \pm 2.3}$ |
| MH-Better-Okay | $156 \pm 50$ | $65.3 \pm 1$ | $39.3 \pm 5.0$ | $\mathbf{82.2 \pm 2.3}$ |
| MH-Okay | $180 \pm 51$ | $54.2 \pm 0$ | $29.8 \pm 4.9$ | $\mathbf{65.1 \pm 2.6}$ |
| MH-Better-Okay-Worse | $194 \pm 93$ | $72.2 \pm 1.2$ | $26.5 \pm 4.4$ | $\mathbf{79.6 \pm 3.6}$ |
| MH-Better-Worse | $201 \pm 107$ | $60.0 \pm 0.8$ | $32.4 \pm 10.7$ | $\mathbf{74.2 \pm 3.3}$ |
| MH-Okay-Worse | $224 \pm 99$ | $52.9 \pm 0.5$ | $28.7 \pm 2.3$ | $\mathbf{67.4 \pm 3.4}$ |
| MH-Worse | $269 \pm 113$ | $52.4 \pm 0$ | $5.8 \pm 4.1$ | $\mathbf{59.5 \pm 6.8}$ |

Table 4: Average training trajectory length and corresponding success rates for DT, CQL, and BC on human-generated datasets in ROBOMIMIC. Each row aggregates over the Lift, Can, and Square tasks.

We see that BC outperformed all other agents. All methods performed best using shortest trajectories and deteriorated similarly as trained on longer trajectories. For full results, refer to Table 13 in Appendix G. The finding that BC performed best is especially interesting in light of Table 3, where BC performed far *worse* than DT. The difference is that Table 3 was on MG (machine-generated) data, while Table 4 studies human-generated data. Orsini et al. (2021) have found the source of data to play a prominent role in determining how well an agent performs on a task. This finding is consistent with many prior works (Brandfonbrener et al., 2022; Mandlekar et al., 2021; Bahl et al., 2022; Gandhi et al., 2022; Wang et al., 2023; Lu et al., 2022) that have found Imitation Learning to work better than Q-Learning when the policy is trained on human-generated data. We corroborate this finding on the Adroit Pen-human-v0 task, also at Table 5.

As trajectory lengths increase, bootstrapping methods are susceptible to worse performance because values must be propagated over longer time horizons. This is especially a challenge in sparse-reward settings and may explain why we observed CQL performing far worse than BC and DT.

Given that we used a context length of 1 for DT in ROBOMIMIC (Section 3.2), the key differences between BC and DT are 1) conditioning on returns-to-go, and 2) the MLP versus Transformer architecture. We hypothesize that BC performs better than DT because the PH and MH datasets

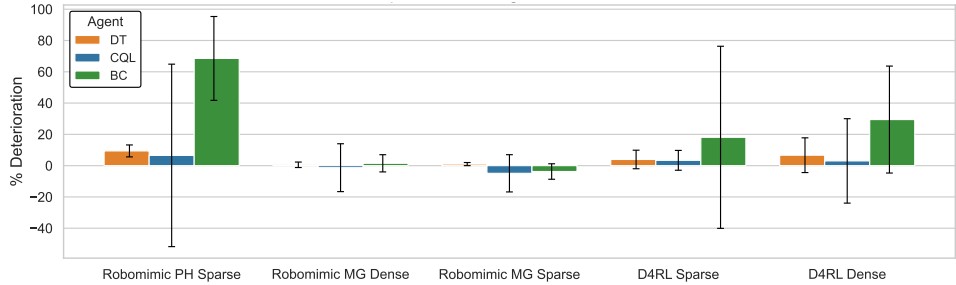

Figure 2: The impact of adding random data to the offline dataset for DT, CQL, and BC. The Y-axis shows the relative deterioration in the evaluation score. Results are averaged over tasks and over two strategies for creating random data. The ROBOMIMIC PH dataset does not have a dense-reward split.

are high-quality enough for Imitation Learning to be effective while being too small for Sequence Modeling. Refer to Appendix F for a detailed study that attempts to disentangle these differences.

> **Practical Takeaway:** Agents are affected similarly as trajectory lengths are increased, but when the data was generated by humans, BC is preferable.

### 4.4 HOW ARE AGENTS AFFECTED WHEN RANDOM DATA IS ADDED TO THE DATASET?

This section explores the impact of adding an equal amount of data collected from a random policy to our training dataset. We considered two strategies to ensure our results were not skewed based on a particular strategy for collecting random data. In "Strategy 1," we rolled out a uniformly random policy from sampled initial states. In "Strategy 2," we rolled out a pre-trained DT agent for several steps and executed a single uniformly random action. The number of steps in each rollout was chosen uniformly at random to be within 1 standard deviation of the average trajectory length in the offline dataset. We can see that Strategy 1 adds random transitions around initial states, while Strategy 2 adds random transitions along the entire trajectory leading to goal states. Because results did not differ significantly between the two strategies (see Appendix G.4), Figure 2 shows results averaged across both. Akin to Section 4.1, we consider dense- and sparse-reward settings in D4RL and ROBOMIMIC.

Compared to BC, both CQL, and DT demonstrated enhanced robustness to injected data, experiencing less than 10% deterioration. However, the resilience of these agents manifests differently. CQL's resilience is more volatile than DT's, as evidenced by the larger standard deviations of blue bars compared to orange ones. In Appendix G.4, we present a per-task breakdown of the results in Figure 2, which shows several intriguing trends. CQL's performance varies greatly across tasks: it improves in some, remains stable in others, and deteriorates in the rest. Interestingly, when CQL's performance on the original dataset is poor, adding random data can occasionally improve it, as corroborated by Figure 1. CQL's volatility is particularly apparent in ROBOMIMIC, with Figure 15 showing that CQL deteriorates nearly 100% on the Lift PH dataset but improves on the Can PH dataset by nearly 2x.

The low deterioration of BC on ROBOMIMIC MG is likely just because the MG data was generated from several checkpoints of training a SAC agent, and therefore already has significantly lower data quality than either ROBOMIMIC PH or D4RL. In Section 4.3, we found BC to be superior when the behavior policy was driven by humans. However, when suboptimal data is mixed with high-quality human data, DT becomes preferable over BC.

> **Practical Takeaway:** BC is likely to suffer the most from the presence of noisy data, while DT and CQL are more robust. However, DT is more reliably performant than CQL in this setting.

### 4.5 HOW ARE AGENTS AFFECTED BY THE COMPLEXITY OF THE TASK?

We now focus on understanding how increasing the task complexity impacts the performance of our agents. Two major factors contributing to task complexity are the dimensionality of the state space and the horizon of the task MDP. To understand the impact of state space dimensionality, we utilized the HUMANOID environment, which has 376-dimensional states, along with other D4RL tasks, which have substantially smaller state spaces. To understand the impact of task horizon, we utilized the ROBOMIMIC PH datasets for Lift, Can, Square, and Transport tasks (listed in order of increasing task

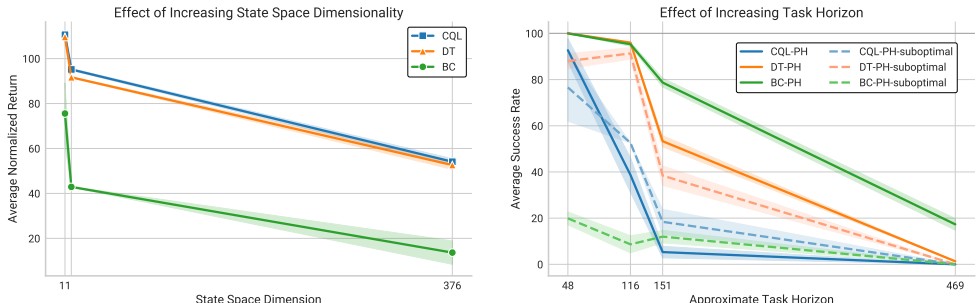

Figure 3: Impact of increasing state-space dimensionality on D4RL medium expert data (left) and task horizon on ROBOMIMIC with both PH data and PH + equal amount of random data (right).

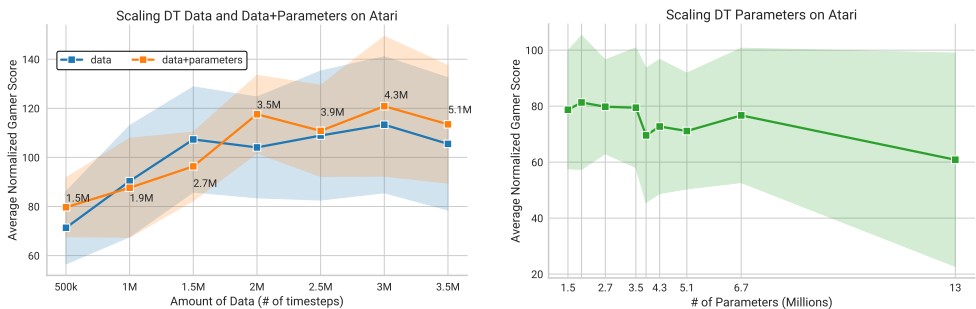

Figure 4: ATARI results for DT while scaling data only (left, blue), parameters only (right), and both simultaneously (left, orange), averaged over BREAKOUT, QBERT, SEAQUEST, and PONG. Numbers on top of the orange curve show model parameter sizes. We found that scaling data was more impactful than scaling parameters, but scaling both together gave minor gains over scaling data only.

horizon). Although trajectory lengths in the dataset are an artifact of the behavior policy while task horizon is an inherent property of the task, we found average trajectory lengths in the dataset to be a useful proxy for quantifying task horizon, because exactly computing task horizon is non-trivial. Akin to the previous section, we experimented with both the PH datasets and the same datasets with an equal amount of random data added in. Figure 3 shows the results averaged on tasks with identical dimensions (left) and task horizon (right) for DT, CQL, and BC.

The performance of all agents degraded roughly equally as the state space dimensionality increased ($11 \rightarrow 17 \rightarrow 376$). Regarding task horizon, with high-quality data (PH), all three agents started near a 100% success rate, but BC deteriorated least quickly, followed by DT and then CQL. In the presence of the suboptimal random data (PH-suboptimal), DT performed the best while BC performed poorly, consistent with our observations in Section 4.4. Additionally, CQL benefited from the addition of suboptimal data, as seen by comparing the solid and dotted blue lines. This suggests that adding such data can improve the performance of CQL in long-horizon tasks, consistent with Figure 1.

> **Practical Takeaway:** All agents experience similar deterioration when the dimensionality of the state space is increased. When the task horizon is increased, DT remains a robust choice, but BC may be preferable when data is known to be high-quality.

### 4.6 HOW DO AGENTS BEHAVE IN STOCHASTIC ENVIRONMENTS ?

Understanding the behavior of these agents in stochastic environment is pivotal for practical applications. We evaluated our previously trained agents under the influence of Gaussian noise added to the predicted action during evaluation, with a probability of $p$ on each timestep as follows: $action = action + (\mathcal{N}(0, 1) * \sigma + \mu)$, where $\sigma$ and $\mu$ are stochasticity parameters. This modification (Fujimoto et al., 2018) is intended to simulate the effect of stochastic environment dynamics. Our findings, presented in Figure 5, reveal that all three agents experience a similar degradation in performance when trained on medium expert data with moderate stochasticity ($p = 0.25, \sigma = 0.25$). However, in the case of sub-optimal (medium replay) data and a higher stochasticity setting ($p = 0.5, \sigma = 0.5$),

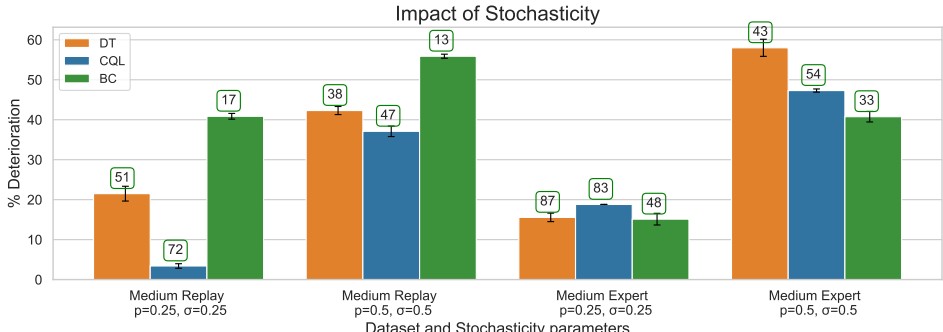

Figure 5: Results on dense-reward D4RL with added stochasticity during evaluation, averaged over the HALFCHEETAH, HOPPER and WALKER tasks. % deterioration (Y-axis) represents the performance drop relative to running evaluations without any stochasticity. Numbers within boxes above each bar represent the average normalized scores obtained on that dataset with the stochasticity parameters.

CQL demonstrated superior resilience compared to DT and BC. Building on the observation made by Paster et al. (2022), who looked at DT's limitations in stochastic environments, our study provides evidence indicating that DT's performance may hold up reasonably well with CQL in continuous action spaces when stochasticity is moderate and data is high-quality.

> **Practical Takeaway:** While DT exhibits a comparable decline to CQL when trained on high-quality data in continuous action spaces, CQL is expected to be relatively more robust as data quality declines or stochasticity increases.

### 4.7 SCALING PROPERTIES OF DECISION TRANSFORMERS ON ATARI

Based on the relative robustness of DT in many of the previous sections, we studied the scaling properties of DT along three dimensions: the amount of training data, the number of model parameters, and both jointly. We focused on the ATARI benchmark because of its high-dimensional observations, for which we expect scaling to be most helpful. We scaled the number of model parameters by increasing the number of layers in the DT architecture $\in \{6, 8, 12, 16, 18, 20, 24, 32, 64\}$. The results are shown in Figure 4. The performance of DT reliably increased with more data up until 1.5M timesteps in the dataset (blue, left), but was insensitive to or perhaps even hurt by increasing the number of parameters (green, right). Upon scaling DT to a certain size (3.5M+ parameters), we observed that the larger model outperformed its smaller counterpart given the same amount of data. Additionally, we discuss architectural properties of DT in Appendix A.

> **Practical Takeaway:** When scaling up DT for complex environments, prioritize scaling the data before the model size. Scaling both simultaneously may improve performance further.

## 5 LIMITATIONS AND FUTURE WORK

This work addressed the question of which learning method amongst CQL, BC and DT should be preferred for offline reinforcement learning. One of the limitations of our work is that we could broaden our study to include more representative algorithms from each paradigm, such as Implicit Q-Learning (Kostrikov et al., 2021) and Trajectory Transformers (Janner et al., 2021), as well as paradigms we did not explore here, such as model-based offline RL (Kidambi et al., 2020b; Yu et al., 2020b) and diffusion models (Ajay et al., 2022). However, we note that resource limitations make this challenging: each datapoint in every plot of our experiments required around 1,000+ GPU-hours, considering the aggregations over domains and random seeds. Each agent we add exponentially increases the computational demand, exceeding our budget. We also hope to evaluate on a larger set of benchmarks that includes compositional tasks, like those in embodied AI (Duan et al., 2022).

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

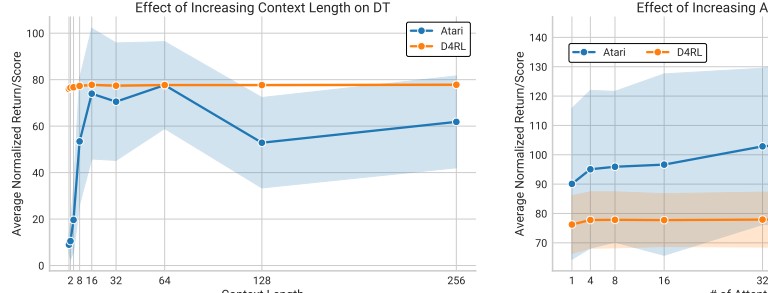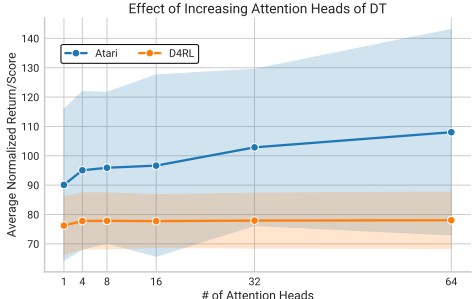

Figure 6: Increasing the context length (left) and number of attention heads (right) of DT on both ATARI and D4RL. ATARI curves average over BREAKOUT, QBERT, SEAQUEST, and PONG. D4RL curves average over all tasks and data splits. We see that scaling context length and attention heads benefits performance on ATARI, but not on D4RL, though the observed benefit of increasing the number of attention heads is not statistically significant.

## A    ARCHITECTURAL PROPERTIES OF DECISION TRANSFORMERS

Here, we study the impact of architectural properties of DT, namely the context length, # of attention heads, # of layers, and embedding size. Full experimental results are in Appendix I.

**Context length:** The use of a context window makes DT dependent on the history of states, actions, and rewards, unlike CQL and BC. Figure 6 (left) demonstrates the role of context length for DT. Having a context length larger than 1 (i.e., no history) did not benefit DT on D4RL, while it helped on ATARI, where performance is maximized at a context length of 64. This finding indicates that some tasks may benefit from a more extensive knowledge of history than others. The deterioration in performance, as context length increases, is likely due to DT overfitting to certain trajectories.

**Attention Heads:** While the importance of the number of Transformer attention heads has been noted in NLP (Michel et al., 2019), it is an open question how the trends carry over to offline RL. Figure 6 (right) shows the impact of this hyperparameter on DT. We observed monotonic improvement on ATARI, but no improvement on D4RL. One of the primary reasons for this discrepancy is that there is more room for an agent to extract higher rewards on ATARI than D4RL. For instance, DT achieved an *expert-normalized* score of over 320 on the BREAKOUT ATARI game, but never gets over 120 in D4RL. This suggests there is more room for scaling the data/parameters to improve results on ATARI.

**Number of layers:** This was discussed in Section 4.7, with results in Figure 4.

**Embedding size:** Increasing the embedding size of DT beyond 256 did not result in any improvement in ATARI; see Table 22 in Appendix I for results.

## B    ADDITIONAL RESULTS

In this section, we provide additional data points on all three agents on different environments of D4RL.

Pen-Human-v0 from Adroit has been generated from human demonstrations and is higher dimensional compared to other D4RL tasks. We observed that DT and BC outperform CQL, while (Brandfonbrener et al., 2022) showed that DT outperformed IQL.

**Trajectory stitching**: Maze environments such as Antmaze and Maze2D require agents to perform trajectory stitching. As (Brandfonbrener et al., 2022) noted, DT based methods require trajectory level information prohibiting DT to make use of information across trajectories. It is due to these reasons that Q-Learning based methods might be considered preferable in the case data requires trajectory stitching to be performed.

| Dataset | DT | CQL | BC |
|---------|-----|------|-----|
| Pen-Human-v0 | $78.36 \pm 4.09$ | $72.52 \pm 2.31$ | $85.68 \pm 6.77$ |

Table 5: Results of all the agents on Pen-Human-v0.

| Dataset | DT | CQL | BC |
|---|---|---|---|
| Antmaze-umaze-v1 | $66.66 \pm 5.31$ | $71.33 \pm 2.05$ | $62.6 \pm 6.8$ |
| Maze 2D Sparse | $6.7 \pm 5.89$ | $19.86 \pm 0.95$ | $15.39 \pm 6.57$ |
| Maze 2D Dense | $27.9 \pm 6.85$ | $49.62 \pm 7.79$ | $38 \pm 5.23$ |

Table 6: Results of all the agents on Maze environments. CQL outperforms DT on these tasks due to the ability of Q learning based methods to perform trajectory stitching.

## C  EXTENDED BACKGROUND

In the reinforcement learning (RL) problem, an agent interacts with a Markov decision process (MDP) (Puterman, 1990), defined as a tuple $(\mathcal{S}, \mathcal{A}, T, R, H)$, where $\mathcal{S}$ and $\mathcal{A}$ denote the state and action spaces, $T(s', a, s) = \Pr(s'|s, a)$ is the transition model, $R(s, a) \in \mathbb{R}$ is the reward function, and $H$ is the (finite) horizon. At each timestep, the agent takes an action $a \in \mathcal{A}$, the environment state transitions according to $T$, and the agent receives a reward according to $R$. The agent does not know $T$ or $R$. Its objective is to obtain a policy $\pi : \mathcal{S} \to \mathcal{A}$ such that acting under the policy maximizes its return, the sum of expected rewards: $\mathbb{E}[\sum_{t=0}^{H} R(s_t, \pi(s_t))]$. In offline RL (Levine et al., 2020), agents cannot interact with the MDP but instead learn from a fixed dataset of transitions $\mathcal{D} = \{(s, a, r, s')\}$, generated from an unknown behavior policy.

**Q-Learning and CQL**. One of the most widely studied learning paradigms is Q-Learning (Sutton and Barto, 2018), which uses temporal difference (TD) updates to estimate the value of taking actions from states via bootstrapping. Although Q-Learning promises to provide a general-purpose decision-making framework, in the offline setting algorithms such as Conservative Q-Learning (CQL) (Kumar et al., 2020) and Implicit Q-Learning (IQL) (Kostrikov et al., 2021) are often unstable and highly sensitive to the choice of hyperparameters (Brandfonbrener et al., 2022). In this paper, we will focus on Conservative Q-Learning (CQL). CQL (Kumar et al., 2020) proposes a modification to the standard Q-learning algorithm to address the overestimation problem by constraining the Q-values so that they do not exceed a lower bound. This constraint is highly effective in the offline setting because it mitigates the distributional shift between the behavior policy and the learned policy. More concretely, CQL adds a regularization term to the standard Bellman update:

$$\min_{\theta} \; \alpha \left( \mathbb{E}_{s \sim \mathcal{D}} \left[ \log \left( \sum_{a'} \exp(Q_\theta(s, a')) \right) \right] - \mathbb{E}_{s, a \sim \mathcal{D}} \left[ Q_\theta(s, a) \right] \right) + \mathsf{TDError}(\theta; \mathcal{D}), \tag{1}$$

where $\alpha$ is the weight given to the first term, and the second term is the distributional $\mathsf{TDError}(\theta; \mathcal{D})$ under the dataset, from C51 (Bellemare et al., 2017).

**Imitation Learning and BC**. When the dataset comes from expert demonstrations or is otherwise near-optimal, researchers often turn to imitation learning algorithms such as Behavior Cloning (BC) or TD3+BC (Fujimoto and Gu, 2021b) to train a policy. BC is a simple imitation learning algorithm that performs supervised learning to map states $s$ in the dataset to their associated actions $a$. Due to the reward-agnostic nature of BC, it typically requires expert demonstrations to learn an effective policy. The clear downside of BC is that the imitator cannot be expected to attain higher performance than was attained in the dataset.

**Sequence Modeling and DT**. Sequence modeling is a recently popularized class of offline RL algorithms that includes the Decision Transformer (Chen et al., 2021) and the Trajectory Transformer (Janner et al., 2021). These algorithms train autoregressive sequence models that map the history of the trajectory to the next action. In the Decision Transformer (DT) model, the learned policy produces action distributions conditioned on trajectory history and desired returns-to-go $\hat{R}_{t'} = \sum_{t=t'}^{T} r_t$. This leads to the following trajectory representation, used for both training and inference: $\tau = (\hat{R}_1, s_1, a_1, \hat{R}_2, s_2, a_2, \ldots, \hat{R}_T, s_T, a_T)$. Conditioning on returns-to-go enables DT to learn effective policies from suboptimal data and produce a range of behaviors during inference.

## D  ADDITIONAL DATASET DETAILS

**Humanoid Data**    In this section, we present the details of the HUMANOID offline Reinforcement Learning (RL) dataset created for our experiments. We trained a Soft Actor-Critic (SAC) agent (Haarnoja et al., 2018) for 3 million steps and selected the best-performing agent, which achieved a score of 5.5k, to generate the expert split. To create the medium split, we utilized an agent that performed at one-third of the expert's performance. We then generated the medium-expert split by concatenating the medium and expert splits. Our implementation is based on (Raffin et al., 2021) and employs the default hyperparameters for the SAC agent (behavior policy). Table 7 displays the performance of all agents across all splits of the HUMANOID task. Furthermore, Table 8 provides statistical information on the HUMANOID dataset.

**Robomimic:**    We visualize the return distribution of ROBOMIMIC tasks, employing a discount factor of 0.99, as illustrated in Figure 7 and Figure 8. It becomes evident that the discount factor has a significant impact on the

| Task | BC | DT | CQL |
|------|-----|-----|-----|
| Humanoid Medium | $13.22 \pm 4.25$ | $23.67 \pm 2.48$ | $\textbf{49.51} \pm \textbf{0.77}$ |
| Humanoid Medium Expert | $13.67 \pm 5.21$ | $52.63 \pm 1.92$ | $\textbf{54.1} \pm \textbf{1.36}$ |
| Humanoid Expert | $24.22 \pm 5.37$ | $53.41 \pm 1.32$ | $\textbf{63.69} \pm \textbf{0.22}$ |
| Average | 17.03 | 43.23 | 55.76 |

Table 7: Performance of agents on a high dimensional state space task. CQL seems better suited to tasks involving higher state space dimensions.

| Task | # of trajectories | # of samples |
|------|-------------------|--------------|
| Humanoid Medium | 4000 | 1.76M |
| Humanoid Expert | 4000 | 3.72M |
| Humanoid Medium Expert | 8000 | 5.48M |

Table 8: Dataset statistics for HUMANOID. The number of samples refers to the number of environment transitions recorded in the dataset.

data's optimality characteristics. PH features shorter trajectories, resulting in a higher proportion of high-return data.

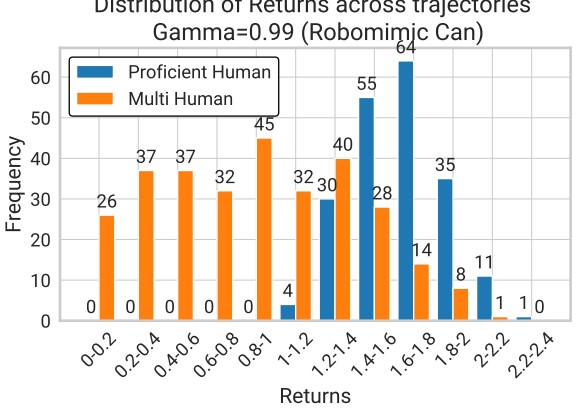

Figure 7: Return Distribution in Proficient and Multi Human splits of Robomimic Can environment shown with gamma set to 0.99. Proficient Human trajectories are shorter in length compared to Multi Human trajectories. Even though PH and MH datasets contain the same number of trajectories, the PH dataset contains more near-expert data.

# E    ADDITIONAL EVALUATION DETAILS

We specify our sampling procedure used to evaluate the ATARI benchmark, used in Section 4.7 and Section A. The ATARI offline dataset (Agarwal et al., 2020) contains the interaction of a DQN agent (Mnih et al., 2015) as it is trained progressively in 50 buffers. Each observation in the dataset contains the last 4 frames of the game, stacked together. Buffers 1 and 50 would contain interactions of the DQN agent when it is naive and expert respectively. Our results are averaged across four different experiments. Each experiment performed a sampling of 500k timestep data from the buffers numbered 1) 1-50 2) 40-50 (DQN is competitive), 3) 45-50, and 4) 49-50 (DQN is expert). We study the architectural and scaling properties of DT in this dataset, where we consider four games: BREAKOUT, QBERT, SEAQUEST, and PONG. We follow the protocol of Lee et al. (2022) and Kumar et al. (2023) by training on the Atari DQN Replay dataset, which used sticky actions, but then evaluating with sticky actions disabled.

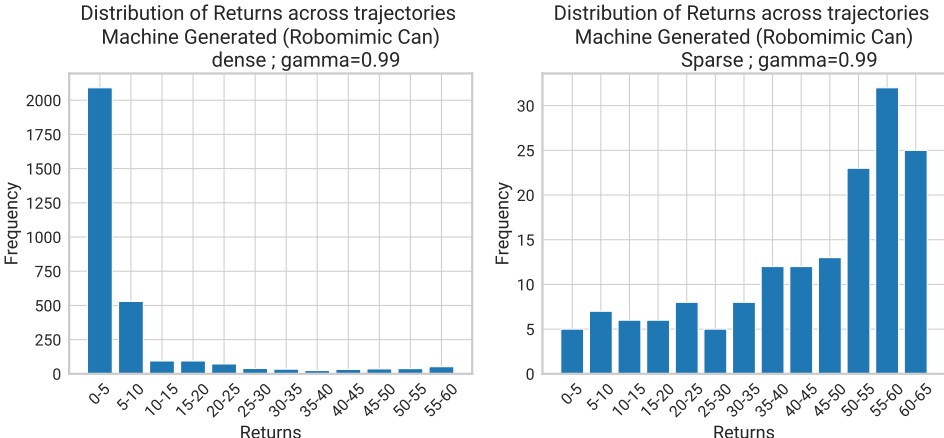

Figure 8: Reward Distribution in Machine Generated splits of Robomimic Can environment is shown with gamma set to 0.99. The sparse variant has more high return trajectories.

## F   DISENTANGLING DT AND BC

In this section, we experimented with an additional baseline, "BC Transformer," a modified version of DT that does not perform conditioning on a returns-to-go vector and has a context length of 1. As previously mentioned, we set the context length to 1 in the case of DT for running experiments on ROBOMIMIC as well. This section aims to investigate the discrepancy between the performance of DT and BC, specifically, we wanted to understand how much of the discrepancy can be attributed to RTG conditioning compared to architectural differences between DT and BC which is typically implemented using a stack of MLPs. By having the BC Transformer baseline, the only distinguishing component between it and BC is the architecture. We observed that BC is often a better-performing agent than BC Transformer on PH Table 9, MG Table 10, and MH tasks Table 11. Additionally, it can also be observed that RTG conditioning only plays a critical role when the distribution of reward shows variation. Unlike expert data such as PH and MH where the RTG vector remained the same, we see DT outperforming BC Transformer on MG significantly.

| Layers | BC TRANSFORMER | DT | BC | CQL |
|---|---|---|---|---|
| Lift-PH | $100 \pm 0$ | $100 \pm 0$ | $100 \pm 0$ | $92.7 \pm 5$ |
| Can-PH | $94.6 \pm 2.4$ | $96 \pm 0$ | $95.3 \pm 0.9$ | $38 \pm 7.5$ |
| Square-PH | $52 \pm 5.8$ | $53.3 \pm 2.4$ | $78.7 \pm 1.9$ | $5.3 \pm 2.5$ |
| Transport-PH | $0 \pm 0$ | $1.3 \pm 0$ | $29.3 \pm 0.9$ | $0 \pm 0$ |

Table 9: Results of agents on ROBOMIMIC PH tasks

| Layers | BC TRANSFORMER | DT | BC | CQL |
|---|---|---|---|---|
| Lift-MG-sparse | $77.9 \pm 7.11$ | $96 \pm 2.1$ | $65.3 \pm 2.5$ | $64 \pm 2.8$ |
| Can-MG-sparse | $48.66 \pm 1.8$ | $92.8 \pm 4.1$ | $64.7 \pm 3.4$ | $1.3 \pm 0.9$ |
| Lift-MG-dense | $68 \pm 3.2$ | $93.99 \pm 2.5$ | $60 \pm 2$ | $63.3 \pm 5.2$ |
| Can-MG-dense | $43.99 \pm 3.2$ | $82.4 \pm 4.2$ | $64 \pm 4.3$ | $0 \pm 0$ |

Table 10: Results of agents on ROBOMIMIC MG tasks

| Layers | BC TRANSFORMER | DT | BC | CQL |
|---|---|---|---|---|
| Lift-MH | $100 \pm 0$ | $100 \pm 0$ | $100 \pm 0$ | $56.7 \pm 40.3$ |
| Can-MH | $93.3 \pm 4.1$ | $95.33 \pm 0$ | $86 \pm 4.3$ | $22 \pm 5.7$ |
| Square-MH | $18.66 \pm 4.1$ | $21.33 \pm 3.7$ | $52.7 \pm 6.6$ | $0.7 \pm 0.9$ |
| Transport-MH | $0 \pm 0$ | $0 \pm 0$ | $11.3 \pm 2.5$ | $0 \pm 0$ |

Table 11: Results of agents on ROBOMIMIC MH tasks

# G   ADDITIONAL RESULTS ON D4RL AND ROBOMIMIC

This section contains results obtained on individual tasks of D4RL and ROBOMIMIC benchmarks. We use the averaged-out results obtained on all of the tasks from the respective benchmark for our analysis.

## G.1   ESTABLISHING BASELINES

This section presents baseline results for individual tasks in both sparse and dense settings of the D4RL. The average outcomes are detailed in Table 2 (Section subsection 4.1). Our observations indicate that DT consistently outperforms CQL and BC in nearly all tasks within the sparse setting of the D4RL benchmark. Although CQL achieves a marginally higher average return on the Hopper task (3.4% ahead of DT for medium and medium-replay splits), it also exhibits significantly higher volatility, as evidenced by the standard deviations. In contrast, DT remains competitive and robust. In the sparse reward setting, CQL surpasses BC by 5.2%. As highlighted in Section subsection 4.1, CQL is most effective in the dense reward setting of the D4RL benchmark.

| Dataset | DT | | CQL | | BC |
| --- | --- | --- | --- | --- | --- |
| | Sparse | Dense | Sparse | Dense | |
| Medium | | | | | |
| Half Cheetah | $\mathbf{42.54 \pm 0.12}$ | $42.55 \pm 0.02$ | $38.63 \pm 0.81$ | $47.03 \pm 0.075$ | $42.76 \pm 0.17$ |
| Hopper | $69.75 \pm 2.31$ | $73.03 \pm 0.74$ | $\mathbf{73.89 \pm 10.12}$ | $70.54 \pm 0.41$ | $64.35 \pm 5.6$ |
| Walker | $\mathbf{75.42 \pm 1.07}$ | $75.42 \pm 0.9$ | $19.31 \pm 3.17$ | $83.77 \pm 0.25$ | $54.62 \pm 12.04$ |
| Average | $62.56 \pm 1.16$ | $63.66 \pm 0.55$ | $43.94 \pm 4.7$ | $67.11 \pm 0.24$ | $53.91 \pm 5.93$ |
| Medium Replay | | | | | |
| Half Cheetah | $\mathbf{38.76 \pm 0.26}$ | $37.09 \pm 0.4$ | $35 \pm 2.56$ | $46.12 \pm 0.06$ | $9.81 \pm 9.2$ |
| Hopper | $82.24 \pm 1.58$ | $89.15 \pm 1.19$ | $\mathbf{83.1 \pm 19.21}$ | $101.27 \pm 0.46$ | $16.19 \pm 10.8$ |
| Walker | $\mathbf{71.24 \pm 1.93}$ | $69.44 \pm 3.14$ | $29.02 \pm 19.63$ | $87.85 \pm 0.83$ | $17.82 \pm 4.96$ |
| Average | $64.08 \pm 1.25$ | $65.22 \pm 1.57$ | $49.04 \pm 13.79$ | $78.41 \pm 0.45$ | $14.6 \pm 8.32$ |
| Medium Expert | | | | | |
| Half Cheetah | $\mathbf{90.55 \pm 1.53}$ | $91.67 \pm 0.21$ | $24.35 \pm 2.38$ | $94.95 \pm 0.25$ | $42.95 \pm 0.14$ |
| Hopper | $\mathbf{110.29 \pm 0.58}$ | $110.76 \pm 0.06$ | $42.44 \pm 12.52$ | $109.67 \pm 2.12$ | $62.21 \pm 6.5$ |
| Walker | $\mathbf{108.63 \pm 0.22}$ | $108.49 \pm 0.1$ | $21.3 \pm 0.55$ | $111.58 \pm 0.17$ | $38 \pm 9.86$ |
| Average | $103.15 \pm 0.77$ | $103.64 \pm 0.12$ | $29.36 \pm 5.14$ | $105.39 \pm 0.84$ | $47.72 \pm 5.5$ |
| Total Average | $\mathbf{76.6 \pm 1}$ | $77.51 \pm 1.12$ | $40.78 \pm 7.88$ | $\mathbf{83.64 \pm 0.51}$ | $38.74 \pm 6.58$ |

Table 12: Results on D4RL in both sparse and dense reward settings, averaged over the HALFCHEETAH, HOPPER, and WALKER tasks. The presented results here are an extension of Table 2.

## G.2 HOW DOES THE AMOUNT AND QUALITY OF DATA AFFECT EACH AGENT'S PERFORMANCE?

Figure 9 illustrates the performance of agents on individual tasks within the D4RL benchmark as data quality and quantity are varied. DT improved or plateaued (upon reaching maximum performance) as additional data was provided. In contrast, CQL exhibits volatility, displaying significant performance declines in HOPPER and WALKER2D medium-replay tasks. The performance of BC tends to deteriorate when trained on low-return data.

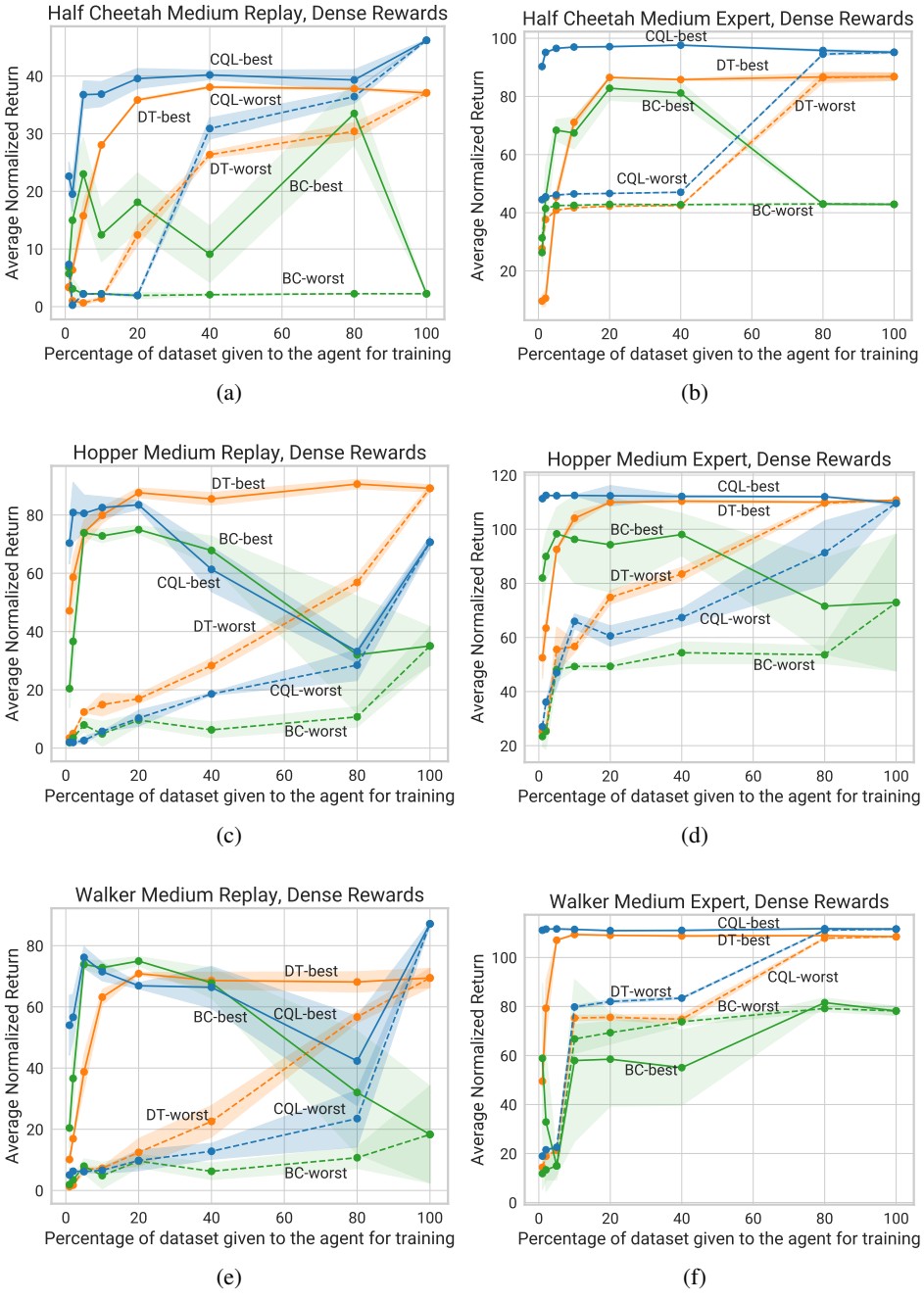

Figure 9: Results demonstrating the behavior of agents as the amount of data and quality is varied on medium-replay and medium-expert splits of D4RL benchmark in dense reward setting. The results presented here are an extension of Figure 1.

Figure 10 illustrates the performance behavior of agents as the quantity and quality of data are adjusted in a sparse setting on the D4RL dataset. A more detailed exploration of this behavior across individual tasks is presented in Figure Figure 11.

Two key observations can be made from these results. 1) In the sparse reward setting, DT becomes a markedly more sample-efficient choice, with its performance either improving or remaining steady as the quantity of data increases. In contrast, CQL displays greater variability and fails to exceed BC in scenarios involving expert data (medium-expert). 2) The sub-optimal data plays a significantly more important role for CQL in sparse settings as compared to dense settings. Our hypothesis is that the sparsity of feedback makes learning about course correction more critical than learning from expert demonstrations. Notably, we discovered that the worst 10% of data features trajectories with substantially higher return coverage, which contributes to greater diversity within the data. This, in turn, enhances CQL's ability to learn superior Q-values (course correction) when compared to the best 10% of data in a medium-expert data setting.

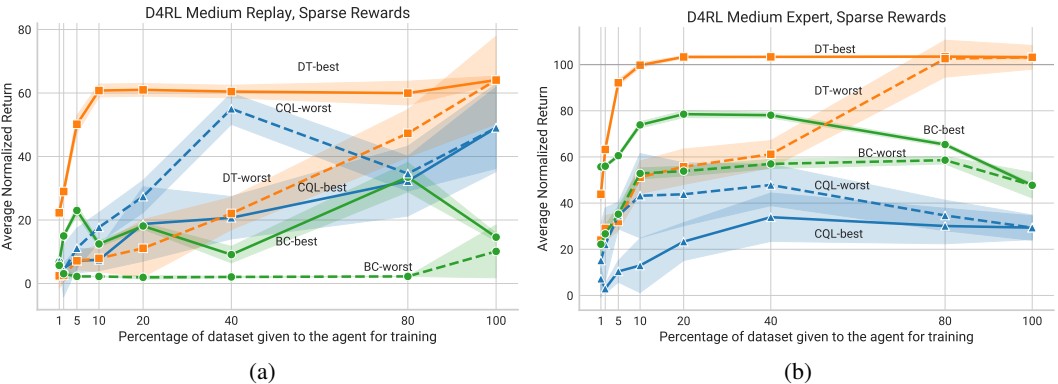

Figure 10: Normalized D4RL returns obtained by training DT, CQL, and BC on various amounts of highest-return ("best") or lowest-return ("worst") data in sparse reward setting. The left plot (a) is on medium replay data, while the right plot (b) is on medium expert data; both plots average over the HALFCHEETAH, HOPPER and WALKER tasks. Notice that the Y-axis limits are set lower on the left plot. We observed that DT was a substantially more sample-efficient agent than CQL. Sub-optimal data was noticeably crucial for CQL in sparse reward setting.

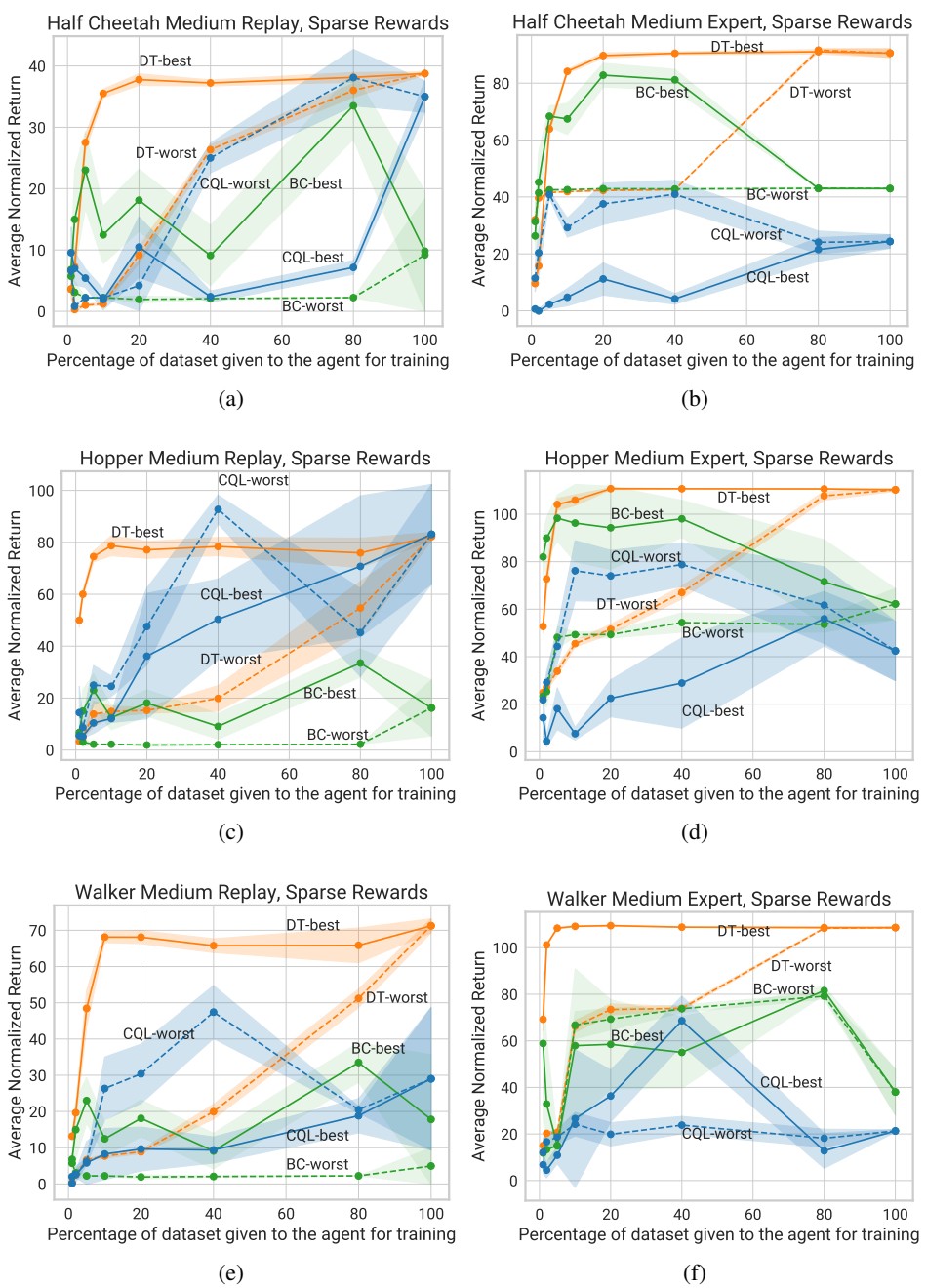

Figure 11: Results demonstrating the behavior of agents as the amount of data and quality is varied on medium-replay and medium-expert splits of D4RL benchmark in sparse reward setting.

### G.3  HOW ARE AGENTS AFFECTED WHEN TRAJECTORY LENGTHS IN THE DATASET INCREASE?

Table 13 showcases the performance of all agents across individual ROBOMIMIC tasks, encompassing both synthetic and human-generated datasets. DT surpasses both agents in all synthetic tasks within the ROBOMIMIC benchmark, for both sparse and dense settings. Interestingly, BC demonstrates a strong aptitude for numerous tasks with human-generated data, particularly excelling on the SQUARE task.

| Dataset Type | Average Trajectory Length | DT | CQL | BC |
|---|---|---|---|---|
| Lift MG Dense | $150 \pm 0$ | $\mathbf{96 \pm 1.2}$ | $68.4 \pm 6.2$ | $59.2 \pm 6.19$ |
| Lift MG Sparse | $150 \pm 0$ | $\mathbf{93.2 \pm 3.2}$ | $60 \pm 13.2$ | $59.2 \pm 6.19$ |
| Can MG Dense | $150 \pm 0$ | $\mathbf{83.2 \pm 0}$ | $2 \pm 1.2$ | $49.6 \pm 2.4$ |
| Can MG Sparse | $150 \pm 0$ | $\mathbf{83.2 \pm 1.59}$ | $0 \pm 0$ | $49.6 \pm 2.4$ |
| Lift-PH | $48 \pm 6$ | $\mathbf{100 \pm 0}$ | $92.7 \pm 5$ | $\mathbf{100 \pm 0}$ |
| Lift-MH-Better | $72 \pm 24$ | $94.6 \pm 1.8$ | $88 \pm 5.9$ | $\mathbf{98.7 \pm 1.9}$ |
| Lift-MH-Okay-Better | $83 \pm 29$ | $100 \pm 0$ | $86 \pm 6.5$ | $\mathbf{99.3 \pm 0.9}$ |
| Lift-MH-Okay | $94 \pm 30$ | $\mathbf{97.33 \pm 0}$ | $67.3 \pm 10.5$ | $\mathbf{96 \pm 1.6}$ |
| Lift-MH | $104 \pm 44$ | $\mathbf{100 \pm 0}$ | $56.7 \pm 5$ | $\mathbf{100 \pm 0}$ |
| Lift-MH-Worse-Better | $109 \pm 49$ | $\mathbf{100 \pm 0}$ | $75.3 \pm 25.6$ | $\mathbf{100 \pm 0}$ |
| Can-PH | $116 \pm 14$ | $\mathbf{96 \pm 0}$ | $38.7 \pm 7.5$ | $95.3 \pm 0.9$ |
| Lift-MH-Worse-Okay | $119 \pm 44$ | $\mathbf{100 \pm 0}$ | $64.7 \pm 2.5$ | $98.7 \pm 1.9$ |
| Can-MH-Better | $143 \pm 29$ | $52.66 \pm 0$ | $20.7 \pm 7.4$ | $\mathbf{83.3 \pm 2.5}$ |
| Lift-MH-Worse | $145 \pm 40$ | $94.6 \pm 0$ | $13.3 \pm 9$ | $\mathbf{100 \pm 0}$ |
| Square-PH | $151 \pm 20$ | $53.3 \pm 2.4$ | $5.3 \pm 2.5$ | $\mathbf{78.7 \pm 1.9}$ |
| Can-MH-Okay-Better | $162 \pm 44$ | $75.33 \pm 3.3$ | $30.7 \pm 7.7$ | $\mathbf{90.7 \pm 1.9}$ |
| Can-MH-Okay | $181 \pm 47$ | $52.66 \pm 0$ | $22 \pm 4.3$ | $\mathbf{72 \pm 2.8}$ |
| Square-MH-Better | $185 \pm 46$ | $13.3 \pm 0$ | $0.7 \pm 0.9$ | $\mathbf{58.7 \pm 2.5}$ |
| Can-MH | $209 \pm 114$ | $\mathbf{95.33 \pm 0}$ | $22 \pm 7.5$ | $86 \pm 4.3$ |
| Can-MH-Worse-Better | $224 \pm 134$ | $\mathbf{73.99 \pm 1.1}$ | $20.7 \pm 5.7$ | $\mathbf{76 \pm 4.3}$ |
| Square-MH-Okay-Better | $225 \pm 76$ | $20.66 \pm 0$ | $1.3 \pm 0.9$ | $\mathbf{56.7 \pm 4.1}$ |
| Can-MH-Worse-Okay | $242 \pm 126$ | $54 \pm 1.6$ | $18.8 \pm 2.5$ | $\mathbf{74.7 \pm 5.7}$ |
| Square-MH-Okay | $265 \pm 78$ | $12.6 \pm 0$ | $0 \pm 0$ | $\mathbf{27.3 \pm 3.4}$ |
| Square-MH | $269 \pm 123$ | $21.33 \pm 3.7$ | $0.7 \pm 0.9$ | $\mathbf{52.7 \pm 6.6}$ |
| Square-MH-Worse-Better | $271 \pm 140$ | $6 \pm 0$ | $1.3 \pm 0.9$ | $\mathbf{46.7 \pm 5.7}$ |
| Can-MH-Worse | $304 \pm 148$ | $51.33 \pm 0$ | $4 \pm 3.3$ | $\mathbf{56.7 \pm 2.5}$ |
| Square-MH-Worse-Okay | $311 \pm 128$ | $4.6 \pm 0$ | $2.7 \pm 1.9$ | $\mathbf{28.7 \pm 2.5}$ |
| Square-MH-Worse | $357 \pm 150$ | $11.3 \pm 0$ | $0 \pm 0$ | $\mathbf{22 \pm 4.3}$ |

Table 13: Success rates for different synthetic and human-generated datasets in ROBOMIMIC. Results presented here are an extension of Table 4 (from subsection 4.3) and have been sorted as per the trajectory length of the task. DT outperformed both agents on synthetic data as seen by performance on MG tasks. BC seemed well suited to numerous tasks for which data was human-generated (PH and MH). Bold numbers represent the best-performing agent on the corresponding task.

### G.4 HOW ARE AGENTS AFFECTED WHEN SUBOPTIMAL DATA IS ADDED TO THE DATASET?

Figure 12 depicts the behavior of agents when random data is introduced according to "Strategy 1" in the dense reward data regime of the D4RL benchmark. As previously described, "Strategy 1" involves rolling out a uniformly random policy from sampled initial states to generate random data. Our observations indicate that both CQL and DT maintain stable performance, while BC exhibits instabilities, occasionally failing as observed in the HALF CHEETAH task.

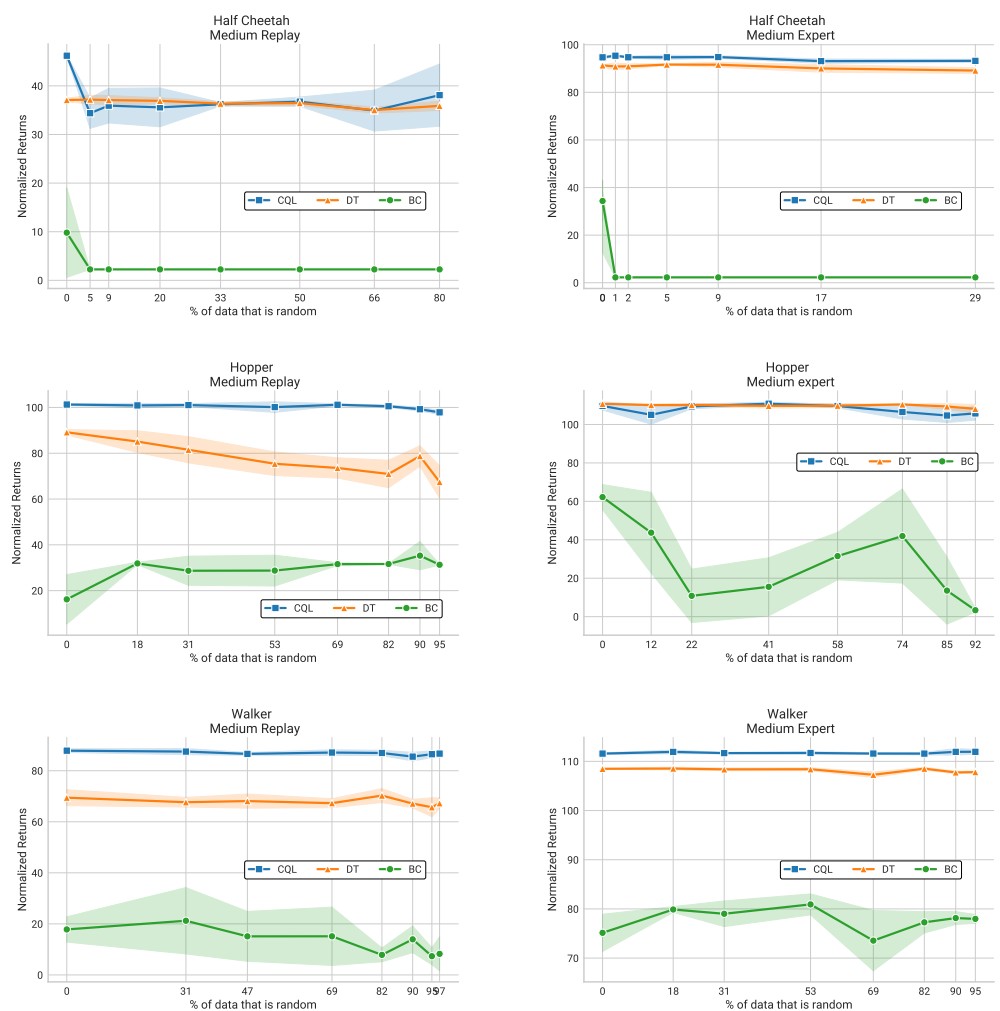

Figure 12: Impact of adding suboptimal data on all agents in dense reward data regime on D4RL tasks using Strategy 1. The results presented here are an extension to Figure 2 (from subsection 4.4).

Figure 13 displays the behavior of agents as random data is incorporated according to Strategy 2 in the dense reward data regime of the D4RL benchmark. In "Strategy 2," we roll out a pre-trained agent for a certain number of steps, execute a single uniformly random action, and then repeat the process. While Strategy 1 produces random transitions primarily clustered around initial states, Strategy 2 generates random transitions across the entire manifold of states, spanning from initial states to high-reward goal states.

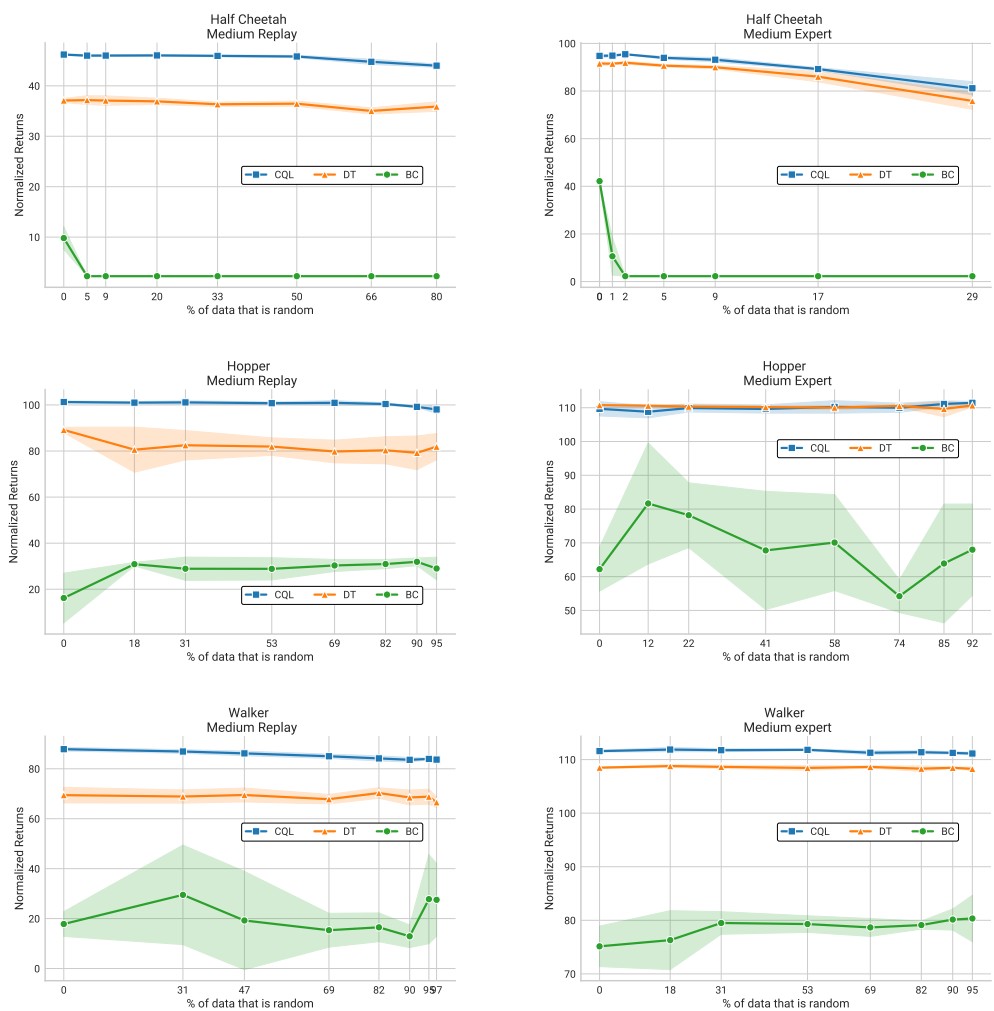

Figure 13: Impact of adding suboptimal data in dense reward data regime on D4RL tasks using Strategy 2. The results presented here are an extension to Figure 2 (from subsection 4.4). A noteworthy observation from this experiment is that the performance of CQL improved on HOPPER medium-expert task with the addition of random data. This suggests that the random data might be aiding in the learning of more accurate Q-values for various states.

Figure 14 illustrates the behavior of agents when random data is introduced according to Strategy 2 in the sparse reward data regime of the D4RL benchmark. We observed that the performance of CQL drastically declined on the HOPPER-MEDIUM-REPLAY task, while its performance stayed the same on other tasks.

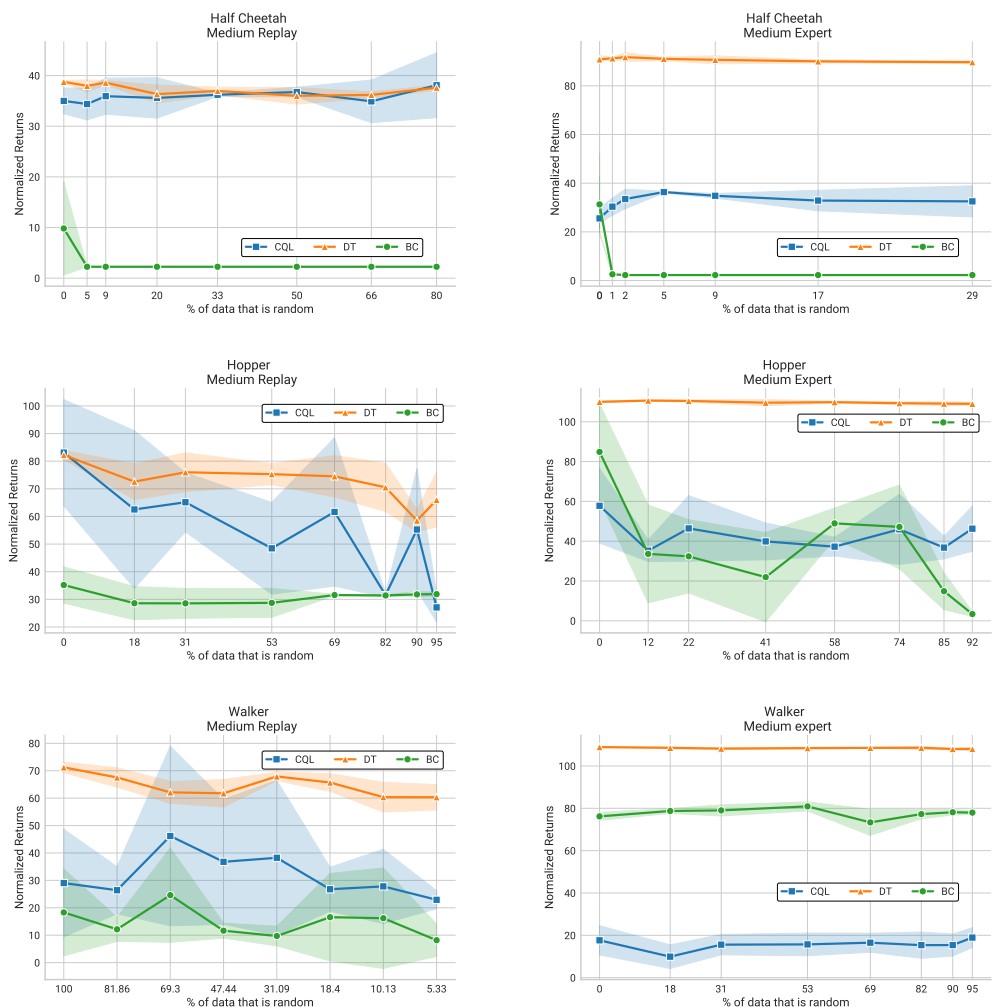

Figure 14: Impact of adding suboptimal data in sparse reward data regime on D4RL tasks using Strategy 2. The results presented here are an extension to Figure 2 (from subsection 4.4). The performance of CQL sharply decreases in the HOPPER medium-replay task. Unlike its performance in the dense reward data regime, CQL does not exhibit the same level of resilience in the sparse reward setting and demonstrates considerably higher volatility.

Figure 15 depicts the behavior of agents when random data is incorporated following Strategy 1 in the sparse reward data regime (human-generated) of the ROBOMIMIC benchmark. Notably, we observed drastically different performance trends with CQL. Its performance plummeted by 80% when the proportion of random data reached 51%. In contrast, its performance nearly doubled on the CAN MG task when the proportion of random data was increased to 30%.

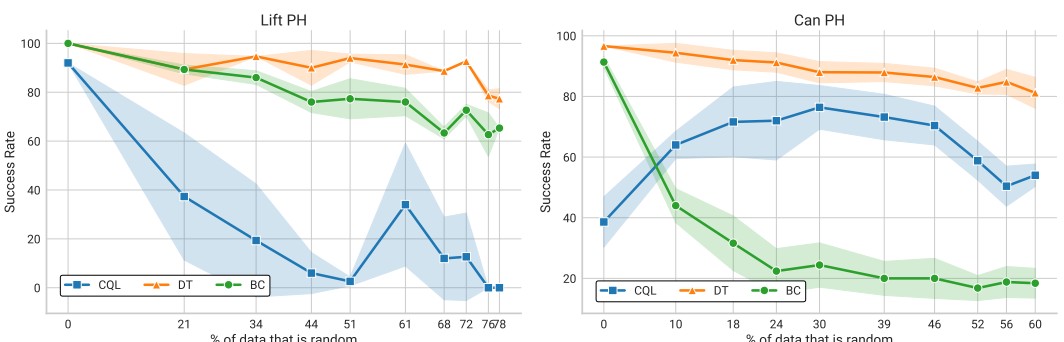

Figure 15: Impact of adding suboptimal data in sparse reward data regime in ROBOMIMIC using Strategy 1. The results presented here are an extension to Figure 2 (from subsection 4.4). CQL exhibited markedly different behaviors across these tasks. Its performance declined sharply on the LIFT PH task. In contrast, its performance significantly improved, nearly doubling, with the addition of random data on CAN PH task.

Figure 16 illustrates the behavior of agents when random data is introduced according to Strategy 2 in the sparse reward data regime (human-generated) of the ROBOMIMIC benchmark.

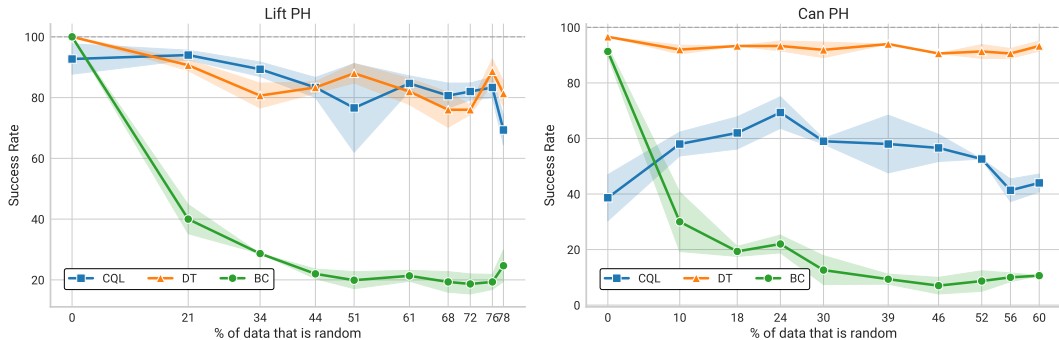

Figure 16: Impact of adding suboptimal data in sparse reward data regime in ROBOMIMIC using Strategy 2. The results presented here are an extension to Figure 2 (from subsection 4.4). Interestingly, when following Strategy 2 for random data generation, CQL maintains stable performance in the LIFT PH task. We observed a similar behavior to that seen with Strategy 1 on the CAN PH task.

Figure 17 and Figure 18 presents the behavior of agents as random data is added following Strategy 2 in sparse and dense reward data regime (synthetic) respectively on the ROBOMIMIC benchmark. In all four scenarios, DT maintained its peak performance reasonably well, indicating its resilience in very noisy data settings. Conservative Q-Learning (CQL), however, showed signs of deterioration on both dense and sparse variants of LIFT tasks. Notably, CQL failed to perform on the CAN task. As mentioned earlier, BC didn't exhibit deterioration, possibly because the Mixed-Goal (MG) data already contains a substantial amount of highly sub-optimal data.

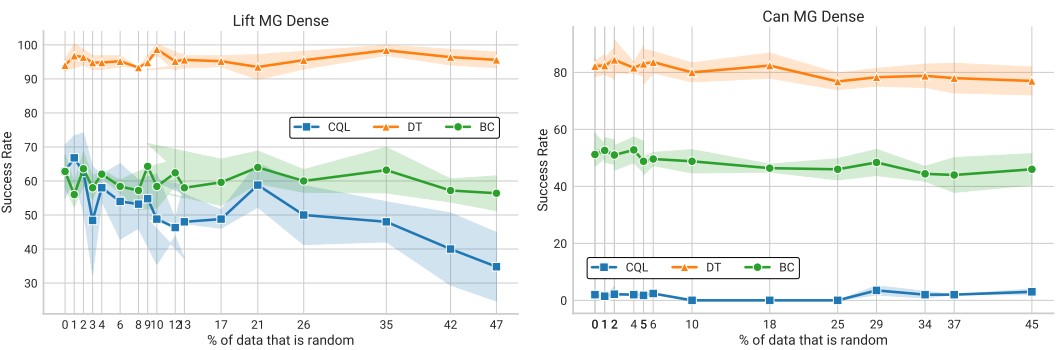

Figure 17: Impact of adding suboptimal data in dense reward data regime in ROBOMIMIC using Strategy 2. The results presented here are an extension to Figure 2 (from subsection 4.4).

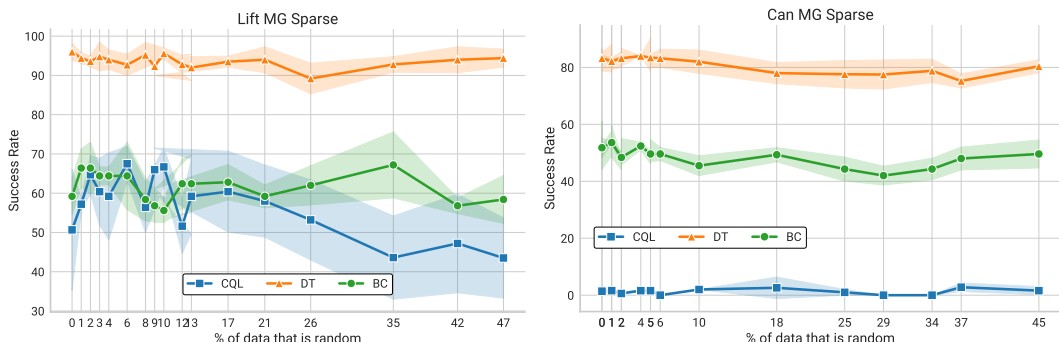

Figure 18: Impact of adding suboptimal data in sparse reward data regime in ROBOMIMIC using Strategy 2. The results presented here are an extension to Figure 2 (from subsection 4.4.

## H  ADDITIONAL EXPERIMENTAL DETAILS

We used the original author implementation as a reference for our experiments wherever applicable. In settings where a new implementation was required, we referenced the implementation which has been known to provide competitive/state-of-the-art results on D4RL. We provide details on compute and hyperparameters below.

**Compute**  All experiments were run on an A100 GPU. Most experiments with DT typically require 10-15 hours of training. Experiments with CQL and BC require 5-10 hours of training. We used Pytorch 1.12 for our implementation.

**Hyperparameters**  We mentioned all the hyperparameters used across various algorithms below. Our implementations are based on original author-provided implementations, without any modifications to the hyperparameters. To learn more about the selection of hyperparameters, we recommend viewing the associated papers. Due to the stable training objective of DT and BC, both of these agents do not require substantial hyperparameter sweep experiments. DT was trained using Adam optimizer (Kingma and Ba, 2014) with a Multi-Step Learning Rate Scheduler. Each experiment was run five times to account for seed variance.

| Hyperparameter | RETURNS-TO-GO |
|---|---|
| Robomimic PH | 6 |
| Robomimic MH | 6 |
| Robomimic MG | 120 |

Table 14: Returns-to-go values used with DT on D4RL and ROBOMIMIC.

| Hyperparameter | VALUE |
|---|---|
| Reference implementation | https://github.com/kzl/decision-transformer/tree/master/atari (MIT License) |
| Number of attention heads | 8 |
| Number of layers | 6 |
| Embedding dimension | 128 |
| Context Length (Breakout) | 30 |
| Context Length (Qbert) | 30 |
| Context Length (Seaquest) | 30 |
| Context Length (Pong) | 50 |
| Number of buffers | 50 |
| Batch size | 16 |
| Learning Rate (LR) | $6e - 4$ |
| Number of Steps | 500000 |
| Return-to-go conditioning | 90 Breakout ($\approx 1\times$ max in dataset) |
| | 2500 Qbert ($\approx 5\times$ max in dataset) |
| | 20 Pong ($\approx 1\times$ max in dataset) |
| | 1450 Seaquest ($\approx 5\times$ max in dataset) |
| Nonlinearity | ReLU, encoder |
| | GeLU, otherwise |
| Encoder channels | $32, 64, 64$ |
| Encoder filter sizes | $8 \times 8, 4 \times 4, 3 \times 3$ |
| Encoder strides | $4, 2, 1$ |
| Max epochs | 5 |
| Dropout | 0.1 |
| Learning rate | $6 * 10^{-4}$ |
| Adam betas | $(0.9, 0.95)$ |
| Grad norm clip | 1.0 |
| Weight decay | 0.1 |
| Learning rate decay | Linear warmup and cosine decay (see code for details) |
| Warmup tokens | $512 * 20$ |
| Final tokens | $2 * 500000 * K$ |

Table 15: Hyperparameters used with DT on ATARI.

| Hyperparameter | VALUE |
|---|---|
| Reference implementation | https://github.com/kzl/decision-transformer/tree/master/gym (MIT License) |
| Number of layers | 3 |
| Number of attention heads | 1 |
| Embedding dimension | 128 |
| Nonlinearity function | ReLU |
| Batch size | 512 |
| Return-to-go conditioning (N/A to BC) | 6000 HalfCheetah |
| | 3600 Hopper |
| | 5000 Walker |
| | 50 Reacher |
| | 6000 Humanoid |
| Dropout | 0.1 |
| Learning rate | $10^{-4}$ |
| Grad norm clip | 0.25 |
| Weight decay | $10^{-4}$ |
| Learning rate decay | Linear warmup for first $10^5$ training steps |
| Context Length (N/A to BC) | 20 |
| Maximum Episode length | 1000 |
| Learning Rate | $6e-4$ |
| Number of workers | 16 |
| Number of evaluation episodes | 100 |
| Number of iterations | 10 |
| Steps per iteration | 5000 |

Table 16: Hyperparameters used with DT and BC on D4RL.

| Hyperparameter | VALUE |
|---|---|
| Reference implementation | https://github.com/tinkoff-ai/CORL/tree/main (Apache 2.0 License) |
| Batch Size | 2048 |
| Steps per Iteration | 1250 |
| Number of Iterations | 100 |
| Discount | 0.99 |
| Alpha multiplier | 1 |
| Policy Learning Rate | 3e-4 |
| QF Learning Rate | 3e-4 |
| Soft target update rate | 5e-3 |
| BC steps | 100k |
| Target update period | 1 |
| CQL n_actions | 10 |
| CQL importance sample | True |
| CQL lagrange | False |
| CQL target action gap | -1 |
| CQL temperature | 1 |
| CQL min q weight | 5 |

Table 17: Hyperparameters used with CQL on D4RL. We used identical hyperparameters as the reference implementation.

| Hyperparameter | VALUE |
|---|---|
| Reference implementation | https://github.com/denisyarats/exorl (MIT License) |
| BC hidden dim | 1024 |
| BC batch size | 1024 |
| BC LR | 1e-4 |
| CQL LR | 1e-4 |
| CQL Critic Target tau | 0.01 |
| CQL Critic Lagrange | False |
| CQL target penalty | 5 |
| CQL Batch size | 1024 |

Table 18: Hyperparameters used with CQL on EXORL. We used identical hyperparameters as the reference implementation.

| Hyperparameter | VALUE |
|---|---|
| Reference implementation | https://github.com/ARISE-Initiative/robomimic (MIT License) |
| BC LR | 1e-4 |
| BC Learning Rate Decay Factor | 0.1 |
| BC encoder layer dimensions | [300, 400] |
| BC decoder layer dimensions | [300, 400] |
| CQL discount | 0.99 |
| CQL Q-network LR | 1e-3 |
| CQL Policy LR | 3e-4 |
| CQL Actor MLP dimensions | [300-400] |
| CQL Lagrange threshold $\tau$ | 5 |

Table 19: Hyperparameters used with BC and CQL on ROBOMIMIC. We used identical hyperparameters as the reference implementation.

# I ABLATION STUDY TO DETERMINE THE IMPORTANCE OF ARCHITECTURAL COMPONENTS OF DT

In this section, we present the results of our ablation study, which was conducted to assess the significance of various architectural components of the DT. To isolate the impact of individual hyperparameters, we altered one at a time while keeping all others constant. Our findings indicate that the ATARI benchmark is better suited for examining scaling trends compared to the D4RL benchmark. This is likely due to the bounded rewards present in D4RL tasks, which may limit the ability to identify meaningful trends. We did not observe any significant patterns in the D4RL context. A key insight from this investigation is that the performance of DT, when averaged across Atari games, improved as we increased the number of attention heads. However, we did not notice a similar trend when scaling the number of layers (Figure 4). It is also important to mention that the original DT study featured two distinct implementations of the architecture. The DT variant used for reporting results on ATARI benchmark had 8 heads and 6 layers, while the one employed for D4RL featured a single head and 3 layers.

| Number of Heads | BREAKOUT | QBERT | SEAQUEST | PONG |
|---|---|---|---|---|
| 4 | $267.5 \pm 97.5$ | $15.4 \pm 11.4$ | $2.5 \pm 0.4$ | $106 \pm 8.1$ |
| 8 | $155.01 \pm 49.86$ | $32.05 \pm 14.23$ | $2.08 \pm 0.44$ | $90.2 \pm 11.33$ |
| 16 | $220.67 \pm 51.02$ | $23.74 \pm 10.75$ | $1.94 \pm 0.39$ | $54.32 \pm 44$ |
| 32 | $246.2 \pm 62.8$ | $34.27 \pm 11.8$ | $1.97 \pm 0.38$ | $72.91 \pm 41.36$ |
| 64 | $249.54 \pm 54.13$ | $24.53 \pm 11.63$ | $2.11 \pm 0.56$ | $69.76 \pm 34.49$ |

Table 20: Ablation study to determine the importance of the number of heads on the performance of DT on the ATARI benchmark. The experiment was conducted with fully mixed Data (50 buffers) with 10 evaluation rollouts (number of layers=6).

| Number of Layers | BREAKOUT | QBERT | SEAQUEST | PONG |
|---|---|---|---|---|
| 6 | $229.58 \pm 49.86$ | $24 \pm 9.28$ | $2.08 \pm 0.44$ | $90.2 \pm 11.33$ |
| 8 | $160.8 \pm 44.1$ | $32.34 \pm 14.07$ | $1.9 \pm 0.36$ | $73.35 \pm 40.4$ |
| 12 | $226.25 \pm 79.47$ | $24.74 \pm 12.61$ | $3.75 \pm 0.78$ | $94.23 \pm 22.44$ |
| 16 | $218.55 \pm 58.92$ | $32.04 \pm 8.49$ | $2.28 \pm 0.59$ | $55.95 \pm 48.55$ |
| 24 | $159.02 \pm 49.35$ | $40.68 \pm 13.99$ | $4.83 \pm 1.3$ | $95.75 \pm 8.36$ |
| 32 | $239.31 \pm 71.21$ | $23.22 \pm 8.15$ | $2.29 \pm 0.46$ | $86 \pm 29.72$ |
| 64 | $223.71 \pm 59.65$ | $31.63 \pm 12.56$ | $2.44 \pm 0.61$ | $73.6 \pm 39.68$ |

Table 21: Ablation study to determine the importance of the number of layers on the performance of DT on the ATARI benchmark. The experiment was conducted with fully mixed Data (50 buffers) with 10 evaluation rollouts (number of heads=8).

| Embedding dimension | BREAKOUT | QBERT | SEAQUEST | PONG |
|---|---|---|---|---|
| 128 | $229.58 \pm 49.86$ | $24 \pm 9.28$ | $2.08 \pm 0.44$ | $90.2 \pm 11.33$ |
| 256 | $289 \pm 76.79$ | $32.34 \pm 14.07$ | $1.91 \pm 0.34$ | $39.25 \pm 48.09$ |
| 512 | $216.17 \pm 49.13$ | $25.59 \pm 9.17$ | - | $2.08 \pm 0.59$ |
| 1024 | $59.7 \pm 78.78$ | $3.93 \pm 8.44$ | $0.11 \pm 0.06$ | - |

Table 22: Ablation study to determine the importance of embedding dimension on the performance of DT on the ATARI benchmark. The experiment was conducted with fully mixed Data (50 buffers) with 10 evaluation rollouts (number of heads=8).

| Context Length | BREAKOUT | QBERT | SEAQUEST | PONG |
|---|---|---|---|---|
| 30 | $229.58 \pm 49.86$ | $24 \pm 9.28$ | $2.08 \pm 0.44$ | $61.04 \pm 38.47$ |
| 32 | $227.51 \pm 68.3$ | $25.27 \pm 8.09$ | $2.16 \pm 0.32$ | - |
| 36 | $213.19 \pm 66.82$ | $23.43 \pm 15.31$ | $2.4 \pm 0.49$ | - |
| 42 | $219.98 \pm 53.5$ | $25.58 \pm 9.86$ | $1.99 \pm 0.35$ | - |
| 48 | $263.29 \pm 49.68$ | $22.91 \pm 15.13$ | $1.96 \pm 0.37$ | - |

Table 23: Ablation study to determine the importance of context length on the performance of DT on the ATARI benchmark. The experiment was conducted with fully mixed Data (50 buffers) with 10 evaluation rollouts (number of heads=8, num of layers=6).

| Heads | BREAKOUT | QBERT | SEAQUEST | PONG |
|---|---|---|---|---|
| 4 | $194.15 \pm 36.66$ | $20.88 \pm 10.05$ | $1.94 \pm 0.37$ | $86.96 \pm 16.67$ |
| 8 | $212.58 \pm 38.55$ | $29.24 \pm 10.83$ | $2.08 \pm 0.31$ | $52.95 \pm 42.54$ |
| 16 | $236.68 \pm 31.73$ | $29.52 \pm 8.37$ | $1.94 \pm 0.3$ | $85.57 \pm 14.29$ |
| 32 | $221.17 \pm 23.81$ | $23.34 \pm 6.34$ | $1.73 \pm 0.24$ | $90.75 \pm 10.99$ |
| 64 | - | $35.55 \pm 10.86$ | $2.01 \pm 0.34$ | $63.36 \pm 36.23$ |

Table 24: Ablation study to determine the importance of heads on DT performance on the ATARI benchmark. The experiment was conducted with fully mixed Data (50 buffers) with 100 evaluation rollouts (number of layers=6).

| Layers | BREAKOUT | QBERT | SEAQUEST | PONG |
|---|---|---|---|---|
| 6 | $212.58 \pm 38.55$ | $29.24 \pm 10.83$ | $2.08 \pm 0.31$ | $70.91 \pm 34.63$ |
| 8 | $220.35 \pm 45$ | $32.42 \pm 13.79$ | $2.12 \pm 0.59$ | $70.4 \pm 36.28$ |
| 12 | $206.83 \pm 46.2$ | $19.48 \pm 7.46$ | $2.06 \pm 0.4$ | $90.77 \pm 12.91$ |
| 16 | $212.67 \pm 36.98$ | $29.04 \pm 10.82$ | $2.22 \pm 0.35$ | $73.84 \pm 37.41$ |
| 24 | $173.43 \pm 56.4$ | $21.41 \pm 10.1$ | $2.11 \pm 0.48$ | $87.5 \pm 15.81$ |
| 32 | $206.07 \pm 45.69$ | $32.15 \pm 9.07$ | $2.18 \pm 0.44$ | $66.4 \pm 40.82$ |
| 64 | $177.64 \pm 102.81$ | $27.74 \pm 5.97$ | $2.04 \pm 0.42$ | $35.98 \pm 43.27$ |

Table 25: Ablation study to determine the importance of layers on the performance of DT on the ATARI benchmark. The experiment was conducted with fully mixed Data (50 buffers) with 100 evaluation rollouts (number of heads=8).

| Heads | BREAKOUT | QBERT | SEAQUEST | PONG |
|---|---|---|---|---|
| 1 | $142.6 \pm 53.85$ | $30.93 \pm 18.4$ | $3.5 \pm 0.81$ | $85.74 \pm 25.64$ |
| 4 | $155.01 \pm 43.58$ | $33.26 \pm 11.12$ | $4.37 \pm 1.25$ | $85.04 \pm 16.68$ |
| 8 | $188.38 \pm 67.43$ | $33.94 \pm 12.46$ | $3.53 \pm 0.73$ | $65.9 \pm 38.08$ |
| 16 | $176.58 \pm 69.9$ | $25.32 \pm 9.71$ | $3.58 \pm 0.96$ | $75.35 \pm 37.8$ |
| 32 | $225.85 \pm 68.21$ | $27.9 \pm 6.47$ | $3.21 \pm 0.73$ | $73.7 \pm 37.27$ |
| 64 | $246.62 \pm 120.95$ | $37.25 \pm 20.03$ | $3.23 \pm 0.87$ | $72.14 \pm 37.78$ |

Table 26: Ablation study to determine the importance of heads on the performance of DT on the ATARI benchmark. The experiment was conducted with the last 10 buffers (out of 50) with 100 evaluation rollouts (number of layers=6).

| Layers | BREAKOUT | QBERT | SEAQUEST | PONG |
|---|---|---|---|---|
| 1 | $151.71 \pm 37.17$ | $21.49 \pm 8.54$ | $3.65 \pm 1.12$ | $79.2 \pm 26.77$ |
| 4 | $186.49 \pm 59.76$ | $28.04 \pm 16.35$ | $3.73 \pm 1$ | $73.55 \pm 37.41$ |
| 6 | $188.38 \pm 67.43$ | $33.94 \pm 12.46$ | $3.53 \pm 0.73$ | $65.9 \pm 38.08$ |
| 8 | $160.8 \pm 44.1$ | $22.52 \pm 14.71$ | $3.31 \pm 1.13$ | $74.11 \pm 37.98$ |
| 12 | $225.85 \pm 68.21$ | $27.9 \pm 6.47$ | $3.21 \pm 0.73$ | $73.7 \pm 37.27$ |
| 16 | $246.62 \pm 120.95$ | $37.25 \pm 20.03$ | $3.23 \pm 0.87$ | $72.14 \pm 37.78$ |
| 24 | $159.02 \pm 49.35$ | $40.68 \pm 13.99$ | $4.83 \pm 1.3$ | $70.4 \pm 35.52$ |

Table 27: Ablation study to determine the importance of layers on the performance of DT on the ATARI benchmark. The experiment was conducted with the last 10 buffers (out of 50) with 100 evaluation rollouts (number of heads=8).

| Heads | HALF CHEETAH | HOPPER | WALKER2D |
|---|---|---|---|
| 4 | 42.74 (0.2) | 77.08 (7) | 74.64 (1.6) |
| 8 | 42.62 (0.2) | 72.3 (3.1) | 74.08 (0.8) |
| 16 | 42.79 (0.2) | 74.07 (3.4) | 73.59 (1.3) |
| 32 | 42.53 (0.2) | 72.76 (1.1) | 76.17 (1.4) |
| 64 | 42.51 (0.1) | 74.14 (1.7) | 74.51 (2.4) |

Table 28: Ablation study to determine the importance of heads on the performance of DT the D4RL medium tasks. The results are averaged across 100 evaluation rollouts (number of layers=6).

| Heads | HALF CHEETAH | HOPPER | WALKER2D |
|---|---|---|---|
| 4 | 37.87 (0.3) | 89.06 (0.4) | 67.93 (2.7) |
| 8 | 38.07 (0.3) | 91.32 (3.6) | 72.7 (2.7) |
| 16 | 38.46 (0.4) | 88.88 (1.3) | 72.16 (1) |
| 32 | 37.88 (0.3) | 88.15 (2.9) | 72.65 (2.8) |
| 64 | 38.17 (0.5) | 90.29 (3.5) | 71.99 (5) |

Table 29: Ablation study to determine the importance of heads on the performance of DT on the D4RL medium-replay tasks. The results are averaged across 100 evaluation rollouts (number of layers=6).

| Heads | HALF CHEETAH | HOPPER | WALKER2D |
|---|---|---|---|
| 4 | 91.52 (0.7) | 110.58 (0.1) | 108.57 (0.3) |
| 8 | 90.55 (0.9) | 110.18 (0.4) | 108.69 (0.3) |
| 16 | 90.61 (0.7) | 110.79 (0.3) | 108.28 (0) |
| 32 | 92.06 (0.2) | 110.84 (0.3) | 108.16 (0) |
| 64 | 91.35 (0.3) | 110.87 (0.4) | 108.45 (0.1) |

Table 30: Ablation study to determine the importance of heads on the performance of DT on the D4RL medium-expert tasks. The results are averaged across 100 evaluation rollouts (number of layers=6).

| Layers | HALF CHEETAH | HOPPER | WALKER2D |
|---|---|---|---|
| 4 | 42.51 (0) | 72.05 (1.8) | 74.68 (0.6) |
| 6 | 42.62 (0.2) | 73.37 (4.4) | 74.08 (0.8) |
| 8 | 42.48 (0) | 73.25 (5.2) | 75.04 (1.9) |
| 12 | 42.55 (0.2) | 70.37 (2.6) | 74.49 (0.89) |
| 16 | 42.45 (0) | 63.3 (1.7) | 74.53 (1.9) |
| 24 | 42.4 (0.1) | 69.07 (4.35) | 75.64 (1.4) |

Table 31: Ablation study to determine the importance of layers on the performance of DT on the D4RL medium tasks. The results are averaged across 100 evaluation rollouts (number of heads=8).

| Layers | HALF CHEETAH | HOPPER | WALKER2D |
|---|---|---|---|
| 4 | 38.8 (0.7) | 92.09 (3) | 69.33 (4.2) |
| 6 | 38.07 (0.3) | 91.32 (3.6) | 72.2 (2.7) |
| 8 | 37.14 (1.3) | 90.71 (0.64) | 70.14 (2.37) |
| 12 | 37.29 (1) | 89.11 (2.1) | 69.36 (2.3) |
| 16 | 37.15 (0.4) | 88.6 (3.3) | 71.4 (1.8) |
| 24 | 37.46 (0.7) | 83.17 (4.6) | 73.66 (2.4) |
| 32 | 38.39 (0.9) | 87.94 (1.6) | 69.43 (0.9) |
| 64 | 38.09 (0.7) | 81.95 (0.46) | 68.01 (2.28) |

Table 32: Ablation study to determine the importance of layers on the performance of DT on the D4RL medium-replay tasks. The results are averaged across 100 evaluation rollouts (number of heads=8).

| Layers | HALF CHEETAH | HOPPER | WALKER2D |
|---|---|---|---|
| 4 | 92.21 (0.2) | 110.26 (0.6) | 108.33 (0.6) |
| 8 | 90.55 (0.9) | 110.18 (0.4) | 108.37 (0) |
| 16 | 90.52 (0.8) | 110.49 (0.2) | 108.79 (0.1) |
| 32 | 90.06 (0.6) | 110.8 (0.2) | 108.61 (0.4) |
| 64 | 90.59 (0.6) | 110.46 (0.4) | 108.82 (0) |

Table 33: Ablation study to determine the importance of layers on the performance of DT on the D4RL medium-expert tasks. The results are averaged across 100 evaluation rollouts (number of heads=8).

| Context Length | HALF CHEETAH | HOPPER | WALKER2D |
|---|---|---|---|
| 10 | 42.68 (0) | 67.6 (3.3) | 75.4 (1.31) |
| 20 | 42.71 (0.1) | 71 (5.4) | 75.25 (0.9) |
| 30 | 42.72 (0.1) | 81.55 (10.2) | 75.8 (0.2) |
| 40 | 42.67 (0.2) | 73.88 (0.9) | 74.73 (1.8) |
| 50 | 42.66 (0) | 83.14 (5.3) | 72.52 (0.7) |
| 60 | 42.49 (0) | 85.54 (6.9) | 74.18 (0.3) |
| 70 | 42.54 (0.1) | 83.75 (0.03) | 74.04 (2.7) |

Table 34: Ablation study to determine the importance of context length on the performance of DT on the D4RL medium tasks. The results are averaged across 100 evaluation rollouts (number of layers=6, number of heads=8).

| Context Length | HALF CHEETAH | HOPPER | WALKER2D |
|---|---|---|---|
| 10 | 36.98 (0.2) | 80.07 (1.1) | 69.73 (2.2) |
| 20 | 37.7 (0.1) | 84.28 (2.6) | 67.62 (2.6) |
| 30 | 35.71 (0.1) | 91.37 (3.9) | 68.71 (1.4) |
| 40 | 35.34 (0.6) | 87.08 (1.5) | 69.33 (2.4) |
| 50 | 35.36 (0.5) | 87.99 (2.3) | 64.61 (5.2) |
| 60 | 37.19 (0.5) | 86.82 (3.5) | 64.55 (1.2) |
| 70 | 36.62 (1.4) | 90.27 (0.8) | 63.48 (2.5) |

Table 35: Ablation study to determine the importance of context length on the performance of DT on the D4RL medium-replay tasks. The results are averaged across 100 evaluation rollouts (number of layers=6, number of heads=8).

| Context Length | HALF CHEETAH | HOPPER | WALKER2D |
|---|---|---|---|
| 10 | 87.61 (2.64) | 109.99 (0.48) | 108.97 (0.92) |
| 20 | 86.21 (3.75) | 109.54 (0.72) | 107.97 (0.82) |
| 30 | 88.24 (2.17) | 110.16 (0.48) | 108.24 (0.26) |
| 40 | 87.38 (1.15) | 110.71 (0.4) | 108.6 (0.04) |
| 50 | 88.22 (0.96) | 110.74 (0.41) | 107.89 (0.56) |

Table 36: Ablation study to determine the importance of context length on the performance of DT on the D4RL medium-expert tasks. The results are averaged across 100 evaluation rollouts (number of layers=6, number of heads=8).

| Context Length | HALF CHEETAH | HOPPER | WALKER2D |
|---|---|---|---|
| 32 | 86.27 (0.3) | 108.92 (0.5) | 108.18 (0.3) |
| 64 | 86.62 (2) | 110.12 (0.7) | 107.85 (0.5) |
| 128 | 89.15 (1.2) | 109.47 (0.3) | 108.53 (0) |
| 256 | 90.98 (0.9) | 109.47 (1.1) | 107.73 (0) |
| 512 | 91.3 (0.3) | 109.61 (0.7) | 107.72 (0) |
| 1024 | 91.38 (0.4) | 109.04 (0.8) | 108.42 (0.2) |

Table 37: Ablation study to determine the importance of context length on the performance of DT on the D4RL medium-replay tasks. The results are averaged across 100 evaluation rollouts (number of layers=6, number of heads=8).

## J DT ON EXORL

We additionally conduct smaller-scale experiments in EXORL, which allows us to study the performance of DT on reward-free play data. Typical offline RL datasets are collected from a behavior policy that aims to optimize some (unknown) reward. Contrary to this practice, the **EXORL** benchmark (Yarats et al., 2022) was obtained from reward-free exploration. After the acquisition of an $(s, a, s')$ dataset, a reward function is chosen and used to include rewards in the data. This same reward function is used during evaluation. We consider the WALKER WALK, WALKER RUN, and WALKER STAND environments (APT). All scores are averaged over 10 evaluation episodes.

In the following section, we present the results obtained using DT on three distinct environments from the EXORL framework. `Returns-to-go` in the tables presented below represent returns-to-value provided to DT at the time of inference. Upon comparing the metrics in the EXORL study, we noticed that DT's performance falls short compared to CQL, which may be attributed to the data being collected in a reward-free setting. Although investigating the behavior of these agents in reward-free settings presents an avenue for future research, we propose the following hypothesis. Typically, the exploration of new states in a reward-free environment is conducted through heuristics such as curiosity (ICM) (Pathak et al., 2017) or entropy maximization (APT) (Liu and Abbeel, 2021). The reward functions defined by these heuristics differ from those used for relabeling data when training offline RL agents. Consequently, bootstrapping-based methods might be better equipped to learn a mapping between the reward function determined by the heuristic and the one employed for data relabeling.

| LR | Returns-to-Go | | | | | | |
|---|---|---|---|---|---|---|---|
| | **200** | **300** | **400** | **500** | **600** | **700** | **800** |
| 6e-4 | 123.28 (1.9) | 120.18 (9.3) | 126.46 (2.1) | 131.82 (7.4) | 128.47 (4.1) | 130.95 (9.1) | 125.61 (7.4) |
| 1e-3 | 119.25 (1.2) | 121.59 (5.1) | 128.85 (5) | 132.35 (6.5) | 136.59 (10.7) | 124.58 (7.7) | 121.68 (5.8) |
| 3e-3 | 123.68 (5.3) | 123.43 (5.3) | 128.23 (0.9) | 122.54 (8.9) | 127.24 (2.2) | 116.33 (1.9) | 122.34 (6.3) |
| 5e-3 | 110.35 (7.3) | 120.65 (8.1) | 123.82 (11.4) | 117.38 (6.7) | 120.45 (2.8) | 112.27 (11.5) | 104.63 (8.1) |
| 7e-3 | 120.6 (7.8) | 125.98 (2.8) | 117.76 (11) | 123.76 (14.9) | 121.7 (6.4) | 110.95 (6.7) | 120.18 (6.9) |
| 9e-3 | 125.04 (2.5) | 114.34 (5.62) | 116.57 (5.8) | 115.17 (1.6) | 123.71 (15.8) | 124.65 (6.8) | 123.9 (5.2) |
| 1-e2 | 126.93 (8.2) | 125.18 (3.9) | 130 (4.2) | 118.92 (10.8) | 107.87 (10.4) | 126.28 (8.7) | 119.57 (13) |

Table 38: Performance of DT on WALKER WALK from EXORL as returns-to-go value is varied. The experiment was run with the following parameters: batch size=7200, warmup steps=10, 200k timestep data, 150 gradient updates, and context length: 20.

| LR | Returns-to-Go | | | | | | |
|---|---|---|---|---|---|---|---|
| | **200** | **300** | **400** | **500** | **600** | **700** | **800** |
| 6e-4 | 133.68 (10.9) | 139.51 (5.1) | 137.5 (12.3) | 131.2 (3.8) | 130.14 (6.5) | 129.8 (1.4) | 129.79 (3.9) |
| 1e-3 | 131.62 (8.8) | 126.34 (5.6) | 125.71 (4.4) | 129.58 (9.2) | 132.2 (11.4) | 127.94 (8.4) | 126.6 (9.5) |
| 3e-3 | 126.71 (3.3) | 129.85 (1.6) | 119.36 (3.7) | 125.56 (1.6) | 128 (12.4) | 125.86 (4.4) | 118.13 (5) |
| 5e-3 | 123.05 (4) | 119.7 (6.5) | 124.79 (7.4) | 124.48 (8.3) | 123.24 (12.3) | 122.25 (8.1) | 129.83 (4.2) |
| 7e-3 | 108.4 (2.5) | 116.85 (6.9) | 120.8 (10.5) | 112.59 (9.1) | 116.12 (3.2) | 113.55 (3.5) | 117.96 (1.8) |
| 9e-3 | 107.79 (7.4) | 98.57 (18) | 110.52 (31.5) | 102.2 (9.6) | 101.21 (20.4) | 102.91 (20.1) | 90.19 (21.9) |
| 1-e2 | 112.69 (10.5) | 100.42 (13.7) | 114 (24.3) | 118.13 (3.6) | 105.55 (9.9) | 108.74 (14.9) | 126.21 (16.2) |

Table 39: Performance of DT on WALKER WALK from EXORL. The experiment was run with the following parameters: batch size=7200, warmup steps=30, 200k timestep data, 150 gradient updates, and context length: 20.

| LR | Warmup | Returns-to-Go | | | | | | |
|---|---|---|---|---|---|---|---|---|
| | | **200** | **300** | **400** | **500** | **600** | **700** | **800** |
| 6e-4 | 30 | 133.68 (10.9) | 139.51 (5.1) | 137.5 (12.3) | 131.2 (3.8) | 130.14 (6.5) | 129.8 (1.4) | 129.79 (3.9) |
| 4e-4 | 30 | 136.16 (4.7) | 131.23 (6.5) | 131.67 (5.2) | 121.06 (6.2) | 121.95 (3.5) | 129.54 (0.1) | 137.26 (0.9) |
| 1e-4 | 30 | 117.7 (4.3) | 124.08 (9.7) | 114.02 (1.9) | 118.9 (3.8) | 119.09 (0.7) | 123.83 (9.7) | 120.86 (7.0) |
| 6e-4 | 50 | 130.14 (11.2) | 129.53 (8.2) | 136.92 (7.4) | 123.96 (0.4) | 132.93 (1) | 123.17 (5.5) | 136.91 (9.6) |
| 6e-4 | 80 | 125.44 (1) | 131.98 (2.3) | 129.48 (7.8) | 123.6 (6.8) | 127.62 (2.9) | 127.64 (6.7) | 123.53 (3.1) |
| 6e-4 | 100 | 124.18 (4.5) | 128.26 (2) | 122.23 (6.8) | 127.9 (0.8) | 125.78 (3.7) | 131.39 (5.9) | 131.83 (5.1) |

Table 40: Performance of DT on WALKER WALK from EXORL. The experiment was run with the following parameters: batch size=7200, 200k timestep data, 150 gradient updates, and a context length of 20.

| LR | Warmup | Returns-to-Go | | | | | | |
|---|---|---|---|---|---|---|---|---|
| | | **200** | **300** | **400** | **500** | **600** | **700** | **800** |
| 6e-4 | 50 | 123.53 (13) | 127.81 (5.3) | 121.44 (6.7) | 112.79 (5.6) | 123.33 (2.7) | 120.44 (1.9) | 124.58 (7) |
| 6e-4 | 100 | 117.02 (4.3) | 137.5 (6.5) | 126.24 (8.1) | 124.64 (6.8) | 125.43 (9.9) | 115.1 (2.0) | 125.67 (7.8) |
| 7e-4 | 100 | 120.91 (9.1) | 128.69 (3.4) | 121.03 (4.4) | 112.11 (6) | 120.67 (4.5) | 127.57 (3.8) | 120.05 (5.9) |
| 1e-3 | 30 | 123.64 (1.3) | 118.84 (11.7) | 129.66 (7.5) | 126.77 (5.5) | 122.59 (7.6) | 127.9 (5.6) | 127.19 (12.3) |

Table 41: Performance of DT on WALKER WALK from EXORL. The experiment was run with the following parameters: batch size=7200, 200k timestep data, 300 gradient updates, and a context length of 20.

| LR | Warmup | Returns-to-Go | | | | | | |
|---|---|---|---|---|---|---|---|---|
| | | **200** | **300** | **400** | **500** | **600** | **700** | **800** |
| 6e-4 | 50 | 57.56 (0.8) | 57.49 (1.4) | 56.25 (1.3) | 56.38 (1.1) | 57.4 (1) | 57.87 (1.6) | 56.23 (2) |
| 6e-4 | 100 | 56.99 (0.6) | 58.64 (0.9) | 57.88 (1.3) | 57.15 (1.3) | 60.11 (2.6) | 57.25 (2.1) | 57.7 (0.8) |
| 8e-4 | 40 | 59.34 (1.8) | 57.98 (2.4) | 57.82 (1.4) | 56.9 (2.2) | 56.48 (2.4) | 60.36 (2.3) | 58.5 (2.1) |
| 6e-4 | 200 | 56.47 (2.5) | 59.69 (2.8) | 56.6 (0.7) | 59.36 (1.0) | 57.33 (2.0) | 59.18 (0.1) | 58.36 (0.1) |
| 7e-4 | 100 | 60.3 (0.1) | 57.12 (0.7) | 57.03 (2.0) | 56.32 (0.7) | 59.55 (1.2) | 58.2 (1.3) | 58.6 (2.2) |
| 1e-3 | 30 | 62.48 (3.3) | 59.23 (1.8) | 56.62 (1.1) | 58.69 (1.7) | 58.94 (2.6) | 56.25 (1.5) | 57.55 (1.7) |

Table 42: Performance of DT on WALKER RUN from EXORL. The experiment was run with the following parameters: batch size=7200, 200k timestep data, 300 gradient updates, context length: 20.

| LR | Warmup | Returns-to-Go | | | | | | |
|---|---|---|---|---|---|---|---|---|
| | | **200** | **300** | **400** | **500** | **600** | **700** | **800** |
| 6e-4 | 100 | 263.71 (6.8) | 292.08 (6.3) | 278.26 (14.7) | 271.19 (3.8) | 283.36 (11.1) | 284.05 (15.2) | 279.87 (8.8) |
| 8e-4 | 40 | 270.83 (1.7) | 271.04 (14.6) | 275.7 (5.5) | 273.92 (13.9) | 269.5 (0.7) | 290.78 (14.7) | 286.77 (6.8) |
| 6e-4 | 200 | 266.37 (14.8) | 271.16 (7.2) | 278.27 (8.2) | 291.59 (9.4) | 287.37 (4.4) | 274.2 (6.9) | 272.6 (5.7) |

Table 43: Performance of DT on WALKER STAND from EXORL. The experiment was run with the following parameters: batch size=7200, 200k timestep data, 300 gradient updates, context length: 20.

| LR | Returns-to-Go | | | | | | |
|---|---|---|---|---|---|---|---|
| | **200** | **300** | **400** | **500** | **600** | **700** | **800** |
| 1e-4 | 128.52 (3) | 125.83 (3.2) | 116.24 (2.5) | 130.23 (12.7) | 137.82 (3.7) | 126.32 (3.4) | 125 (5.5) |
| 6e-4 | 114.56 (11.9) | 107.41 (5.5) | 127.49 (10.4) | 115.08 (8.4) | 99.1 (4.0) | 117.1 (7.3) | 97.73 (1.3) |
| 9e-4 | 106.24 (7.5) | 102.84 (16.0) | 102.74 (14.7) | 95.55 (5.5) | 110.13 (10.4) | 105.28 (13.3) | 100.03 (7.0) |
| 1e-3 | 92.71 (3.1) | 104.86 (15.3) | 99.35 (1.5) | 96.8 (8.0) | 86.95 (10.0) | 98.01 (14.7) | 92.7 (1.6) |
| 4e-3 | 45.65 | 42.86 | 39.6 | 41.93 | 39.71 | 39.88 | 43.05 |
| 8e-3 | 56.3 | 39.3 | 44.63 | 48.93 | 39.35 | 44.12 | 52.88 |

Table 44: Performance of DT on WALKER WALK from EXORL. The experiment was run with the following parameters: batch size=7200, warmup steps=100, 200k timestep data, 900 gradient updates.

| LR | Returns-to-Go | | | | | | |
|---|---|---|---|---|---|---|---|
| | **200** | **300** | **400** | **500** | **600** | **700** | **800** |
| 1e-4 | 113.53 (2.5) | 123.63 (2.7) | 115.69 (2.0) | 116.66 (1.7) | 114.54 (1.1) | 115.48 (2.1) | 122.86 (6.4) |
| 6e-4 | 103.02 (5.5) | 102.17 (1.3) | 94.05 (0.6) | 98.45 (0.0) | 95.22 (1.5) | 101.86 (8.9) | 114.71 (3.8) |
| 9e-4 | 92.71 (8.7) | 92.18 (0.8) | 102.49 (5.0) | 88.61 (11.8) | 92.95 (3.2) | 94.97 (2.2) | 97.31 (3.6) |
| 1e-3 | 84.65 | 93 | 97.18 | 93.57 | 95.27 | 92.04 | 88.7 |
| 4e-3 | 57.96 (1.0) | 66.73 (2.9) | 61.93 (14.4) | 46.71 (3.4) | 58.77 (9.0) | 51.75 (0.2) | 47.44 (5.1) |
| 8e-3 | 39.59 (2.4) | 38.9 (4.6) | 44.71 (12.2) | 41.27 (2.9) | 40.34 (5.8) | 48.63 (4.7) | 45.2 (7.2) |

Table 45: Performance of DT on WALKER WALK from EXORL. The experiment was run with the following parameters: batch size=7200, 200k timestep data, warmup steps=100, 900 gradient updates, context length=10.

| LR | Returns-to-Go | | | | | | |
|---|---|---|---|---|---|---|---|
| | **200** | **300** | **400** | **500** | **600** | **700** | **800** |
| 1e-4 | 113.53 (2.5) | 123.63 (2.7) | 115.69 (2.0) | 116.66 (1.7) | 114.54 (1.1) | 115.48 (2.1) | 122.86 (6.4) |
| 6e-4 | 103.02 (5.5) | 102.17 (1.3) | 94.05 (0.6) | 98.45 (0.0) | 95.22 (1.5) | 101.86 (8.9) | 114.71 (3.8) |
| 9e-4 | 92.71 (8.7) | 92.18 (0.8) | 102.49 (5.0) | 88.61 (11.8) | 92.95 (3.2) | 94.97 (2.2) | 97.31 (3.6) |
| 1e-3 | 84.65 | 93 | 97.18 | 93.57 | 95.27 | 92.04 | 88.7 |
| 4e-3 | 57.96 (1.0) | 66.73 (2.9) | 61.93 (14.4) | 46.71 (3.4) | 58.77 (9.0) | 51.75 (0.2) | 47.44 (5.1) |
| 8e-3 | 39.59 (2.4) | 38.9 (4.6) | 44.71 (12.2) | 41.27 (2.9) | 40.34 (5.8) | 48.63 (4.7) | 45.2 (7.2) |

Table 46: Performance of DT on WALKER WALK from EXORL. The experiment was run with the following parameters: batch size=7200, 200k timestep data, warmup steps=100, 900 gradient updates, context length=10.

| LR | Returns-to-Go | | | | | | |
|---|---|---|---|---|---|---|---|
| | **200** | **300** | **400** | **500** | **600** | **700** | **800** |
| 1e-4 | 116.61 (7.8) | 111.97 (6.3) | 114.26 (4.4) | 116.88 (4.9) | 123.97 (3.4) | 111.21 (3.7) | 108.62 (2.5) |
| 6e-4 | 112.06 (6.3) | 113.17 (2.5) | 104.78 (3.3) | 112.57 (6.7) | 107.5 (3.4) | 112.69 (4.6) | 106.85 (2.8) |
| 9e-4 | 108.42 (5.7) | 102.44 (2.3) | 99.46 (8.1) | 105.42 (1.7) | 106.62 (9.3) | 99.2 (6.2) | 102.55 (8.4) |
| 1e-3 | 103.74 (6.5) | 106.42 (1.8) | 98.7 (6.2) | 103.34 (4.2) | 116.92 (3.4) | 107.22 (7.0) | 105.08 (11.6) |
| 4e-3 | 87.2 (6.4) | 83.95 (4.2) | 89.53 (8.1) | 77.35 (12.4) | 90.34 (2.6) | 89.65 (8) | 93.22 (7.8) |
| 8e-3 | 57.73 (10.6) | 59.11 (9.3) | 56.23 (4.2) | 58.82 (12.9) | 52.2 (3.6) | 62.79 (2.4) | 54.29 (9.0) |

Table 47: Performance of DT on WALKER WALK (Proto) from EXORL. The experiment was run with the following parameters: batch size=7200, 200k timestep data, 900 gradient updates, context length=10.

| LR | Returns-to-Go | | | | | | |
|---|---|---|---|---|---|---|---|
| | **200** | **300** | **400** | **500** | **600** | **700** | **800** |
| 1e-4 | 116.61 (7.8) | 111.97 (6.3) | 114.26 (4.4) | 116.88 (4.9) | 123.97 (3.4 ) | 111.21 (3.7) | 108.62 (2.5) |
| 6e-4 | 112.06 (6.36) | 113.17 (2.5) | 104.78 (3.3) | 112.57 (6.7) | 107.5 (3.4) | 112.69 (4.6) | 106.85 (2.8) |
| 9e-4 | 108.42 (5.7) | 102.44 (2.3) | 99.46 (8.1) | 105.42 (1.7) | 106.62 (9.3) | 99.2 (6.2) | 102.55 (8.4) |
| 1e-3 | 103.74 (6.5) | 106.42 (1.8) | 98.7 (6.2) | 103.34 (4.2) | 116.92 (3.4) | 107.22 (7.0) | 105.08 (11.6) |
| 4e-3 | 87.2 (6.4) | 83.95 (4.2) | 89.53 (8.1) | 77.35 (12.4) | 90.34 (2.6) | 89.65 (8.0) | 93.22 (7.8) |
| 8e-3 | 57.73 (10.6) | 59.11 (9.3) | 56.23 (4.2) | 58.82 (12.9) | 52.2 (3.6) | 62.79 (2.4) | 54.29 (9.0) |

Table 48: Performance of DT on WALKER WALK (Proto) from EXORL. The experiment was run with the following parameters: batch size=7200, 200k timestep data, 900 gradient updates, context length=10.

