# OpenReview forum: "When should we prefer Decision Transformers for Offline Reinforcement Learning?"
_ICLR.cc/2024/Conference — ICLR 2024 poster_

### Official Review · Reviewer_wytN · 2023-10-27

**Soundness:** 4 excellent
**Presentation:** 4 excellent
**Contribution:** 3 good
**Rating:** 6
**Confidence:** 4

**Summary:**

The authors compare three approaches to Offline-RL: CQL (representing Q-learning approaches), BC (IL approaches), and Decision Transformer (Sequence Modeling approaches) under a range of environments and data regimes (e.g. varying the amount of data, quality of data, or whether it was generated by a human or agent).
Based on the results, they distill concise, actionable advice for practitioners.

**Strengths:**

I think this paper is a useful read for every practitioner in Offline-RL as it concisely evaluates and tests a range of relevant environmental factors and how those should influence the choice of algorithm.
Fairly comparing and evaluating different algorithms against each other is a seemingly simple, but actually not that straightforward thing to do as a lot of choices have to be made well: For example, which algorithms to compare, which environments to use and how to distill the results into concise statements that readers will actually remember.
And I think the authors did a great job here in this regard.
Of course, there are various additional things that they could have looked at (e.g. continuous vs. discrete actions) or a more modern approach than BC - but I do understand that lines have to be drawn somewhere.

**Weaknesses:**

I think the paper has no important weaknesses w.r.t what it set out to do, namely comparing important algorithmic approaches on a range of relevant environments.
In terms of publication, one could argue that there is limited novelty - not just in terms of algorithmic novelty, but also insights generated: None of the insights were very surprising and some were quite obvious (e.g. CQL performs worse when the back-propagation of the reward signal is impeded through sub-optimal or longer trajectories).

I do believe this is a valid concern, but nevertheless regard the paper as very useful for the community, hence my initial rating of 6 (with a tendency towards 8).

**Questions:**

Please let me know if I mischaracterized or overlooked some of your contributions.

---

> ### Author Response · Authors · 2023-11-17
>
> We are very grateful for the reviewer's feedback.
>
> We have highlighted how our work allows one to determine when DT would be preferred in offline RL in the common response section. We would like to highlight that many insights that are valuable to the community are not necessarily surprising (for ex. Scaling model size and data). All of our insights are backed up by thorough empirical analysis, providing readers with actionable and reliable insights. We emphasized more on the continuous action space environment primarily due to its importance over discrete action space. Having more experiments required more time, resources which might lead to diminishing returns or not significantly alter our conclusions. We hope you would understand, and want to urge the reviewer to consider increasing the score.

---

> > ### Comment · Reviewer_wytN · 2023-11-22
> >
> > Thank you for your reply!

---

### Official Review · Reviewer_q2Sb · 2023-10-29

**Soundness:** 3 good
**Presentation:** 3 good
**Contribution:** 3 good
**Rating:** 6
**Confidence:** 4

**Summary:**

The paper investigates three offline RL algorithms DT, CQL and BC under different settings, to see the effective use case for each algorithm. The experiments involve the settings with sparse-dense rewards, different data qualities, various trajectory lengths, random data injection, different complexity of tasks, stochastic environments and scaling with data and model parameters. The experiments are quite thorough with observed facts summarized, as well as conjectural reasons. It is an interesting paper providing some insights for the offline RL community.

**Strengths:**

The paper is well-written with small formatting issues. The experiments are quite thorough.

In comparing DT, CQL and BC, the paper involves at least seven different aspects. The observed facts are quite informative. Apart from summarizing the observed facts, the authors also provide reasonable explanations for most of the results.

**Weaknesses:**

Although explanations are provided for most of the observed facts, most of them are speculative without further evidence. For example, in Sec. 4.1, ``a likely reason … mean CQL must propagate TD errors …’’. If these conjectures are dived deeper with further evidence and justification, it will make the paper more solid.

Another major critique is that the algorithms CQL and BC are no longer the SOTA results for these offline RL baselines, and more effective algorithms are proposed with improvement. Examples like TD3BC [1], percentage BC, Diffusion Q-learning [2] for BC type methods,  and implicit Q-learning, implicit diffusion Q-learning [3], for Q-learning type methods, etc. Comparing DT with CQL, BC is a good starting point, but it could also be more interesting to see those more advanced algorithms in these comparisons. For example, it is known that BC does not perform well as long as the dataset contains not only high-return samples, but how does percentage BC perform in settings other than Sec. 4.2? Would it be a robust and reliable method as well in the offline setting?

Minor issues:

Position of Tab. 2 is problematic.

Reference:

[1] Fujimoto, Scott, and Shixiang Shane Gu. "A minimalist approach to offline reinforcement learning." Advances in neural information processing systems 34 (2021): 20132-20145.

[2] Wang, Zhendong, Jonathan J. Hunt, and Mingyuan Zhou. "Diffusion policies as an expressive policy class for offline reinforcement learning." arXiv preprint arXiv:2208.06193 (2022).

[3] Hansen-Estruch, Philippe, et al. "Idql: Implicit q-learning as an actor-critic method with diffusion policies." arXiv preprint arXiv:2304.10573 (2023).

**Questions:**

In Fig. 1(a), it shows when the dataset contains more sub-optimal samples, CQL degrades its performance more than DT, but when it uses 100% data the performance increases back to as good as DT. Is there any explanation for this?

It is interesting to see that DT is more reliable than CQL when adding random data (as Sec. 4.4), while less robust when injecting stochasticity in the environment (as Sec. 4.6). What’s the potential reason for DT to perform less robustly in the latter case?

---

> ### Author Response · Authors · 2023-11-17
>
> We would like to express our gratitude to the reviewer for providing us with thorough feedback.
>
> 1. We have addressed the explanation concern in Common Response section
> 2. We appreciate the reviewers' suggestions to consider additional methods like IQL and TD3BC etc. We do not have a consensus on which algorithms can be considered SOTA in offline RL literature due to many confounding factors (stochasticity, dense/sparse settings etc.). For the sake of keeping our work relevant and broadly applicable, it is reasonable to consider algorithms which have become mainstream and are widely used. Our choice of algorithms (CQL, DT, BC) was driven by their widespread acceptance, ease of interpretation and compute budget. The simplicity of these algorithms, combined with our extensive understanding of them as a community, allows us to clearly trace performance fluctuations, which is crucial for our study's comprehensive analysis. In this way, our work provides valuable insights that enhance understanding of offline RL agents. We would like to highlight the compute and time concerns from the Common Response section to emphasize why having these agents made the most sense.
>
> Due to these concerns, it was reasonable to leave out the diffusion based approach (it is yet to be widely adopted).
>
> Here are some additional reasons why we did not study other methods.
> - Numerous papers that are pivotal and investigative in nature (Engstrom et al. 2019, Andrychowicz et al. 2021, Ilyas et al. 2019) have showcased results on a small set of algorithms and environments:
> Brandfonbrener et al. 2022 showed empirical results on BC, IQL and DT on 1 toy environment and 3 D4RL environments with standard configuration.
> Kumar et al. 2022 showed results on only CQL and BC on 3 D4RL environments with standard configuration.
> - GAIL (Ho et al. 2016) requires online reinforcement learning, diverging from our chosen pure offline RL setting where data remains static. TD3BC adds the BC objective into CQL. BC regularization is not as good as the one from CQL. When we are dealing with low-return data, we know it leans more on the BC objective. It is less interpretable.
> - IQL could potentially bolster the perceived strength of our results. However, there's no consensus on its superiority over CQL. In many scenarios, CQL outperforms IQL (Kostrikov et al. 2021).
> - Trajectory Transformer (Janner et al. 2021) employs planning in the learned model and is model based, unlike the model free methods which are the focus of our study. Additionally GATO  (Reed et al. 2022) is not open sourced, making DT the most appropriate choice in the paradigm.
> - We know that %BC only considers a small subsample of the existing dataset. It throws out potentially useful data. For making a fairer comparison, it made sense for each of the agents to see the same amount of data.

---

> > ### Comment · Reviewer_q2Sb · 2023-11-22
> >
> > Thanks for response.
> >
> > I think my questions about in-depth analysis beyond speculative reasons for observations are not addressed.
> >
> > For percentage BC, I may disagree that it throws out potentially useful data. For BC approach, the quality of the data could be more essential than the amount of it. So I would expect to see the results of percentage BC from the paper.
> >
> > Given above, although I generally like the perspectives taken by the paper, I cannot raise my score for now. The paper provides some insights for the domain but can definitely be enhanced to be a stronger one.

---

### Official Review · Reviewer_2vu5 · 2023-10-31

**Soundness:** 3 good
**Presentation:** 4 excellent
**Contribution:** 2 fair
**Rating:** 6
**Confidence:** 4

**Summary:**

The paper empirically investigates the performance of three representative offline RL algorithms – Conservative Q-learning (CQL), Decision Transformer (DT), and Behavior Cloning (BC) – as several dimensions of the dataset, task, or model are varied, with the aim of determining which algorithm is most appropriate, depending on the setting. The authors provide concrete conclusions and recommendations based on their findings.

**Strengths:**

* The experiments investigate a number of significant practical considerations in offline RL, such as reward structure, inclusion of suboptimal data, and task complexity.
* The authors provide concrete conclusions and recommendations based on their findings. A practitioner trying to decide which algorithm to use for their task/dataset would likely benefit from reading this paper.
* The paper is clear, readable, and presented well.
* The D4RL-style humanoid dataset may be of independent interest to researchers, and I hope that the authors will release it.

**Weaknesses:**

* I am skeptical of the sparsified version of D4RL because it violates the Markov property of the reward function: the reward now depends on the whole trajectory rather than just the current state/action. And two trajectories may end at the same state but have significantly different total rewards, causing disagreement. I can see why this transformation would break CQL but not DT, and indeed CQL deterioriates much more on sparse D4RL than sparse Robomimic. So I think the results here do not convincingly show that CQL is worse than DT for sparse environments in general; on D4RL, this artificial transformation violates an assumption underlying the CQL algorithm, and on Robomimic, CQL was already worse than DT on Robomimic even when given dense rewards.
* The experiment on stochasticity does not seem to perfectly answer the question “how good are these algorithms in stochastic environments?” because the data you are training on was collected with deterministic dynamics. So really what you are testing is “how well does a policy trained by these algorithms in a deterministic MDP transfer to a similar stochastic MDP?”. This may also be an interesting question to study, but I think that if you intend to answer the first question, the experiment is not entirely appropriate, and it would be best to repeat it with the training data also collected in a stochastic MDP.
* Some of the conclusions regarding BC are obvious (e.g. BC suffers more than offline RL when there is suboptimal/noisy data) or already widely known (e.g. BC works better than offline RL on human-generated demonstrations). It could provide more insight to include a hybrid method such as TD3-BC and show how the degree of regularization affects the behavior.

**Questions:**

* Am I misunderstanding something in my complaints about the D4RL sparsification or the stochasticity? Please correct me if so.
* What is a “high-quality continuous action space”? (near top of pg. 9) Does this just refer to the degree of determinism?

---

> ### Author Response · Authors · 2023-11-17
>
> We appreciate the efforts the reviewer took to provide feedback. We address all of the concerns of the Reviewer below.
>
> 1) **Sparse setting in a Markovian environment**.
> We agree that the existing D4RL sparse setting might not favor CQL due to non Markovian dynamics (similar to original DT study), we ran experimentation on Maze2D dense and split variants. Additionally we have experiments on Robomimic which also follows Markovian dynamics to back up our claims.
>
> The following experiment was run on the sparse and dense variants of Maze as provided by the D4RL authors. It is worth noting that the Maze task explicitly requires trajectory stitching. As noted by Brandfonbrenfer et al. 2021, sequence modeling based approach would struggle in trajectory stitching. We observe this empirically here. Below we provide all the results sparse vs dense in addition to the experiments in paper to make comparison easier.
>
> |                     | CQL           | DT (RTG=400)  | BC            |
> |---------------------|---------------|---------------|---------------|
> | Maze 2D Sparse      | $19.86(0.95)$ | $6.7(5.89)$   | $15.39(6.57)$ |
> | Maze 2D Dense       | $49.62(7.79)$ | $27.9(6.85)$  | $38(5.23)$    |
> | D4RL Sparse         | $83.64(0.51)$ | $103.15(0.77)$| $38.74(6.58)$ |
> | D4RL Dense          | $40.78(7.88)$ | $76.6(1)$     | $38.74(6.58)$ |
> | Robomimic Sparse    | $30(6.6)$     | $88.2(1.6)$   | $57.2(6)$     |
> | Robomimic Dense     | $35.2(3.7)$   | $89.6(1.4)$   | $57.2(6)$     |
> -----
> We will adjust the claim to highlight that DT struggles on tasks requiring trajectory stitching and CQL should be a preferred choice. Due to shortage of time, we did not tune any parameters (incl. Experimenting with RTGs)
>
> 2) **Stochastic setting**.
> For comparison with existing stochastic experiments provided in the paper, we trained all the agents on deterministic data from D4RL, and introduced stochasticity during inference on stochastic (both params kept to 0.25).
>
> |                   | CQL        | DT (RTG=400) | BC       |
> |-------------------|------------|--------------|----------|
> | Maze 2D Sparse    | 21.54(2.2) | 8.5(2.35)    | 4.62(3.91)|
> | Maze 2D Dense     | 48.5(5.64) | 37.79(3.76)  | 33.72(5.4)|
>
> Here also we see the trend where CQL emerges to be a preferred choice. The claims made in the paper remain intact.
>
> One important concern which was raised is, we did not train agents in a stochastic setting. This is not a common setting in offline RL as the existing datasets are typically obtained in a deterministic environment. To address this concern, we trained a SAC agent to generate our own Maze 2D offline RL data in the same way as we generated humanoid with stochasticity parameters both set to 0.1 (prob=0.1, sigma=0.1) (trained for 2x longer to account for instability).
>
> The following results are obtained when all three agents were trained on stochastic data followed by inference in stochastic setting (both params set to 0.25)
>
> |               | CQL         | DT (RTG=400) | BC        |
> |---------------|-------------|--------------|-----------|
> | Maze 2D Dense | 30.71(0.04) | 36.31(4.58)  | 32.4(6.88)|
>
> We observed that DT outperforms CQL and BC in this scenario. This is an interesting finding showing that when all three agents are trained in a stochastic setting, CQL may or may not be the best choice when evaluated in a stochastic setting.
>
> 3) While some of the conclusions regarding BC were known to the community (such as it typically thrives in expert data regime), our work provided multitudes of additional insights about BC which are novel (are not widely known or have not been shown empirically with this much thoroughness), we mention here a few points that can be considered novel about BC.
> - The trend shown in Figure 1 and Figure 9 of %BC gives a deep insight about the behavior of DT as more high return and low return data is made more accessible, at a much more granular level not seen previously in other works.
> - BC can be a preferred choice over CQL in sparse environments (especially in cases where rewards are not known)
> - BC shows impressive performance on offline RL tasks, outperforming DT and CQL, when data is collected from human demonstrations. We have observed this extensively and have been observed in previous works.
> - BC stays competitive when the task horizon is increased compared to other agents.
>
> The main point we want to highlight is all of the claims are backed up by rigorous experimentation, providing researchers with reliable insights.
>
> 4) By “high-quality continuous action space”, we are pointing out that if the offline RL data contains trajectories which are deemed high quality (medium-expert), then the drop in performance of DT in stochastic settings might be comparable to that of CQL as our findings show. We will rephrase this sentence to make it more clear.
>
> We hope this addresses the concerns raised by Reviewer 2vu5 and humbly request them to reconsider the score.

---

> > ### Comment · Reviewer_2vu5 · 2023-11-22
> > **Response to authors**
> >
> > Thank you for conducting additional experiments in response to my concerns! I would say that they have been addressed, and assuming the paper is edited in accordance with your new results and conclusions, I have raised my score.

---

### Official Review · Reviewer_pkaE · 2023-11-06

**Soundness:** 3 good
**Presentation:** 3 good
**Contribution:** 2 fair
**Rating:** 6
**Confidence:** 2

**Summary:**

This paper contributes to the field of offline Reinforcement Learning (RL) by conducting a comprehensive research study comparing three popular algorithms: CQL, BC, and DT. It investigates their performance across benchmark datasets like D4RL and ROBOMIMIC, analyzing their behavior under varying data quality, task complexity, and stochasticity.

The key findings reveal that DT requires more data than CQL to achieve competitive performance, but exhibits higher robustness to suboptimal data. Notably, DT outperforms both CQL and BC in sparse reward and low-quality data settings. Interestingly, DT and BC demonstrate superior performance for tasks with longer horizons or data collected from human demonstrations. Additionally, CQL shines in situations characterized by both high stochasticity and lower data quality.

Beyond comparative analysis, the paper delves deeper into DT, exploring optimal architectural choices and scaling trends on ATARI and D4RL datasets. Notably, it demonstrates that increasing the data volume by fivefold results in a 2.5x average score improvement for DT on ATARI.

Overall, this research offers valuable insights for selecting the most appropriate offline RL algorithm based on the specific characteristics of the task and data conditions.

**Strengths:**

This paper conducts a thorough empirical analysis on offline RL algorithms, focusing on DT and the comparison to other offline RL algorithms, in particular, CQL, and BC. The paper presents a number of valuable practical insights, which offer great value to the RL community regarding selecting offline RL algorithms in different scenarios.

The paper is well-organized, especially listing the main practical insights at the end of each section, which makes readers easy to follow and understand.

**Weaknesses:**

1. While the paper offers great empirical analysis, it would be still great to provide some technical/theoretical justifications regarding each empirical result shown in the paper, similar to [1], which will further strengthen the claim in the paper.

2. The empirical experiments are done in Robomimic and D4RL, but the D4RL tasks are only mujoco locomotion tasks. The authors should include experiments on more complex tasks such as Androit, Antmaze and vision-based tasks, e.g. vision-based drawer manipulation tasks in [1] and Atari games.

[1] Kumar, Aviral, Joey Hong, Anikait Singh, and Sergey Levine. "Should i run offline reinforcement learning or behavioral cloning?." In International Conference on Learning Representations. 2021.

**Questions:**

1. Please offer theoretical insights in each of the empirical findings where DT is more robust.
2. Please perform experiments in more complex tasks such as Androit, Antmaze and vision-based tasks, e.g. vision-based drawer manipulation tasks in [1] and Atari games.

---

> ### Author Response · Authors · 2023-11-17
>
> We thank the reviewer for taking time to provide feedback.
>
> We addressed the theoretical justification concern in the common response section.
>
> 2) As requested by the reviewer, we are providing additional results on more complex tasks of D4RL such as Antmaze and Adroit. We have tons of results on Atari in Appendix (please see Section H) (due to lack of space in main text space). We were not able to add vision tasks due to reasons mentioned in the Common Response section.
>
> |                   | CQL        | DT (RTG=400) | BC      |
> |-------------------|------------|--------------|---------|
> | Antmaze-umaze-v1  | 71.33(2.05)| 66.66(5.31)  | 62.6(6.8)|
> | Pen-Human-v0      | 72.52(2.31)| 78.36(4.09)  | 85.68(6.77)|
>
> - Antmaze also requires agents to explicitly stitch trajectories. Based on the empirical findings on this environment, we will highlight the trajectory stitching drawback with DT.
> - Results on Adroit corroborate our finding that BC is a superior choice when the data comes from human demonstrations followed by DT. A trend we have seen previously with Robomimic.
>
> We hope this addresses all of the concerns of Reviewer pkaE.

---

### Author Response · Authors · 2023-11-17
**Common Response**

We are grateful to all reviewers for taking the time to review our manuscript, and we appreciate that the reviewers recognized our study as thorough, valuable, insightful and organized.

Our primary goal with this paper is to offer an encompassing perspective on different paradigms in offline RL, given the recent traction gained by Decision Transformers (DT). We considered algorithms from each of the paradigms that have been broadly used (mainstream), interpretable and are competitive. The key differentiator of our paper versus other offline RL studies is that we framed a broad spectrum of research questions which required modifying existing datasets to reflect realistic challenges commonly encountered by researchers in a very rigorous manner.  While existing works (such as Kumar et al. 2022, Brandfonbrener et al. 2022) leaned heavily on theoretical analysis, concretely showcasing where DT and BC fall short against Q-learning on existing off-the-shelf datasets, our paper provides novel insights to answer where DT might be considered preferable over other agents with thorough and wide ranging empirical evaluations on more challenging and practical environments (long horizon, varying definitions of suboptimal data (human/synthetic), varying noise/state space in data, stochasticity) etc., validating some previously shown hypotheses and bringing new insights. This makes our study the first of its kind in exploring aspects of offline RL scenarios that previous literature has yet to address, thereby offering actionable insights into the potential of each agent. To appropriately reflect the contributions of the paper, we are also willing to change our paper title to “When should we prefer Decision Transformers for offline reinforcement learning?".

One common concern that was raised by multiple reviewers was more theoretical justification for empirical results. However, the focus of our work was to investigate how theoretical intuitions for various classes of methods play out in practice across a diverse set of problems. We had to determine which agents, environments along with the questions we were asking should be prioritized. We considered D4RL, Robomimic, Atari and exoRL (only mentioned in the appendix since it's collected in a reward free manner). We rigorously tested each experiment, running for 500k to 1M+ timesteps across all D4RL and Robomimic environments (6 environments) with all three agents on five random seeds. This, along with parameter variations (% data that is optimal, % of random data etc.), demanded substantial computational resources (1200+ GPU hours per data point, Fig 1 left plot required 10k+ GPU hours). Each agent, environment or experiment we add exponentially increases the computational demand, exceeding our budget. Considering the resources, time and current depth of our analysis, we believe we have reached a significant point where further experiments might lead to diminishing results or significantly extend the timeline without substantially altering the conclusions (as reviewer wytN mentioned, lines have to be drawn somewhere). Therefore, we prioritized thoroughness of the questions we sought to address over the breadth of methods, given our constraints.

We will adjust the claims wherever required, include the experiments and make discussion (during rebuttal) part of the paper, wherever possible upon acceptance.

---

### Author Response · Authors · 2023-11-22

As a gentle reminder, discussion ends today. If you have any additional comments, please let us know so we have time to respond. If you feel like your feedback and our responses together have made the paper stronger, we would very much appreciate you updating your score.

---

### Meta-Review · Area_Chair_8v2N · 2023-12-06

**Metareview:**

This paper presents an empirical study on three popular offline RL algorithms---Conservative Q-Learning (CQL), Behaviour Cloning (BC), and Decision Transformer (DT). The paper exposes the three algorithms to multiple challenging scenarios such as sparse rewards, lower-quality training data, long horizon and large stochasticity, and find that each algorithm shines in specific scenarios, with DT being an overall robust choice.

All reviewers find the empirical results to be well executed and extremely useful for practitioners, with concerns about a "lack of novelty" per se as well as some of the explanations being speculative. Given the brittleness of deep RL practice, I think the such findings are quite timely and valuable, and outweigh the concerns. Therefore, I recommend acceptance.

**Justification For Why Not Higher Score:**

The paper mostly reports empirical findings about existing methods with some explanations, with a lacking of more in-depth analyses or rigorous justifications.

**Justification For Why Not Lower Score:**

The findings presented in this paper are thorough and provide many valuable insights for RL practice.

---

### Decision · Program_Chairs · 2024-01-16

Accept (poster)